# ER-resident sensor PERK is essential for mitochondrial thermogenesis in brown adipose tissue

Hironori Kato[1], Kohki Okabe[2], Masato Miyake[3], Kazuki Hattori[4], Tomohiro Fukaya[5], Kousuke Tanimoto[6], Shi Beini[2], Mariko Mizuguchi[7], Satoru Torii[8], Satoko Arakawa[8], Masaya Ono[9], Yusuke Saito[10], Takashi Sugiyama[1], Takashi Funatsu[2], Katsuaki Sato[5], Shigeomi Shimizu[8], Seiichi Oyadomari[3], Hidenori Ichijo[4], Hisae Kadowaki[1], Hideki Nishitoh[1]

**Mitochondria play a central role in the function of brown adipocytes (BAs). Although mitochondrial biogenesis, which is indispensable for thermogenesis, is regulated by coordination between nuclear DNA transcription and mitochondrial DNA transcription, the molecular mechanisms of mitochondrial development during BA differentiation are largely unknown. Here, we show the importance of the ER-resident sensor PKR-like ER kinase (PERK) in the mitochondrial thermogenesis of brown adipose tissue. During BA differentiation, PERK is physiologically phosphorylated independently of the ER stress. This PERK phosphorylation induces transcriptional activation by GA-binding protein transcription factor α subunit (GABPα), which is required for mitochondrial inner membrane protein biogenesis, and this novel role of PERK is involved in maintaining the body temperatures of mice during cold exposure. Our findings demonstrate that mitochondrial development regulated by the PERK–GABPα axis is indispensable for thermogenesis in brown adipose tissue.**

## Introduction

Brown adipose tissue (BAT) is one of the major tissues causing non-shivering thermogenesis in homeothermic animals exposed to cold stress and plays an important role in metabolic function that contributes to energy consumption (Cannon & Nedergaard, 2004). Thermogenesis in brown adipocytes (BAs) is mediated by the function of uncoupling protein 1 (UCP1), which localizes to the mitochondrial inner membrane and dissipates the mitochondrial proton electrochemical gradient (Susulic et al, 1995; Matthias et al, 2000; Feldmann et al, 2009). The development of BAs consists of two

steps: lineage commitment from precursor cells to brown pre-adipocytes and differentiation from brown preadipocytes into mature BAs (Harms & Seale, 2013; Kajimura & Saito, 2014). Differentiated BAs have unique morphological characteristics; these cells possess multiple lipid droplets (LDs) and a number of expanded mitochondria that contain dense parallel cristae (Napolitano & Fawcett, 1958). The highly developed cristae are effective in maintaining the mitochondrial membrane potential (ΔΨm), which is essential for two main functions: oxidative phosphorylation (OXPHOS)–dependent ATP production, which mainly occurs in LD-associated mitochondria, and thermogenesis mediated by cytoplasmic-free mitochondria (Benador et al, 2018). However, the mechanism by which BAs acquire these developed mitochondria remains unknown.

Some areas on the mitochondrial surface make close contact with the ER membrane in various types of cells (Kato & Nishitoh, 2015). ER–mitochondria contact dynamically fluctuates in response to various types of stimuli and regulates a number of cellular functions, such as calcium homeostasis (Rizzuto et al, 1998; Hirabayashi et al, 2017), lipid biosynthesis (Kornmann et al, 2009), mitochondrial dynamics regulated by fusion and fission (Friedman et al, 2011), and autophagy (Hamasaki et al, 2013). Although the ER in differentiated BAs is not as developed as it is in other secretory cells, a large area of the ER membrane in BAs attaches to the mitochondrial outer membrane (de Meis et al, 2010; Golic et al, 2014), and ER-resident molecules contribute to mitochondrial biogenesis (Bartelt et al, 2018; Zeng et al, 2019). However, the molecular mechanism by which ER–mitochondria crosstalk regulates the functions of BAs remains unclear.

In mammalian cells, three types of ER-resident stress sensors, PKR-like ER kinase (PERK), inositol-requiring enzyme 1α (IRE1α), and activating transcription factor (ATF) 6, are activated by ER stress, resulting in activation of the unfolded protein response (UPR). Under ER stress conditions, activation of PERK is triggered by the dissociation of glucose-regulated protein (GRP) 78 (also known as

[1]Laboratory of Biochemistry and Molecular Biology, Department of Medical Sciences, University of Miyazaki, Miyazaki, Japan [2]Laboratory of Bioanalytical Chemistry, Graduate School of Pharmaceutical Sciences, The University of Tokyo, Tokyo, Japan [3]Division of Molecular Biology, Institute for Genome Research, Institute of Advanced Medical Sciences, Tokushima University, Tokushima, Japan [4]Laboratory of Cell Signaling, Graduate School of Pharmaceutical Sciences, The University of Tokyo, Tokyo, Japan [5]Division of Immunology, Department of Infectious Diseases, Faculty of Medicine, University of Miyazaki, Miyazaki, Japan [6]Genome Laboratory, Medical Research Institute, Tokyo Medical and Dental University (TMDU), Tokyo, Japan [7]Department of Immunology, Graduate School of Medicine, University of the Ryukyus, Okinawa, Japan [8]Department of Pathological Cell Biology, Medical Research Institute, TMDU, Tokyo, Japan [9]Department of Clinical Proteomics, National Cancer Center Research Institute, Tokyo, Japan [10]Division of Pediatrics, Faculty of Medicine, University of Miyazaki, Miyazaki, Japan

Correspondence: nishitoh@med.miyazaki-u.ac.jp

BiP) from its luminal domain, followed by oligomerization and auto-phosphorylation. Activated PERK phosphorylates eukaryotic translation initiation factor 2 subunit α (eIF2α), leading to attenuation of global protein translation to reduce the ER load (Harding et al, 2000). Phosphorylation of eIF2α triggers the specific translation of ATF4, which activates the transcription of genes involved in the UPR. Although PERK enrichment in the mitochondria-associated ER membrane (MAM) has been reported to contribute to ER stress-induced apoptosis (Verfaillie et al, 2012; Rainbolt et al, 2013; Lebeau et al, 2018), whether PERK regulates mitochondrial homeostasis in BAs is largely unknown. Mitochondrial biogenesis is regulated by coordination between mitochondrial DNA transcription and nuclear DNA transcription activated by several transcription factors, including Nrf-1, Sp1, YY-1, ERRs, TFAM, and GA-binding protein transcription factor α subunit (GABPα) (Dorn et al, 2015). The mechanisms by which these transcription factors strictly recognize mitochondrial conditions and are activated to regulate mitochondrial biogenesis are unclear.

Here, we show a novel function of PERK in BAs that is independent of the UPR. PERK is phosphorylated at the kinase insert region, presumably by a non-autophosphorylation mechanism, during BA differentiation. PERK is required for mitochondrial and thermogenic gene expression via transcriptional activation by GABPα and UCP1-mediated thermogenesis in vitro and in vivo. Overall, our data suggest that the activation of the PERK–GABPα pathway during BA differentiation is indispensable for mitochondrial inner membrane protein biogenesis and thermogenesis in BAT.

# Results

## Development of mitochondria and increases in ER–mitochondria contact sites during BA differentiation

The morphology of organelles, especially mitochondria, in BAT has been well characterized for more than half a century (Lever, 1957; Napolitano & Fawcett, 1958). Recent reports have revealed that ER-localized molecules contribute to mitochondrial biogenesis and function in BAT (Bartelt et al, 2018; Zeng et al, 2019). We first investigated the morphological changes in mitochondria and the ER in BAs during differentiation. Primary brown preadipocytes were isolated from the interscapular BAT (iBAT) of newborn mice and differentiated into BAs, and cells were harvested at each stage of differentiation (Fig 1A). A few LDs started to be observed on day 4, and multiple LDs emerged on day 6 (Fig S1), suggesting that adipogenesis was nearly completed on day 6. The mitochondrial area and perimeter in differentiating and differentiated BAs (days 2, 4, and 6) were greater than those in brown preadipocytes (day 0) (Fig 1B and C). In contrast with mitochondria, the expanded ER areas and perimeters in brown preadipocytes (day 0) were substantially and significantly reduced during differentiation (Fig 1B and D). OXPHOS-dependent ATP production was calculated by measurement of total and oligomycin A–insensitive intracellular ATP levels. During differentiation, the total amount of intracellular ATP was reduced (Fig 1E), whereas the ratio of OXPHOS-dependent ATP production to total ATP content was significantly increased (Fig 1F).

Moreover, we measured the oxygen consumption rate (OCR) and found that not only the basal OCR but also the extracellular acidification rate (ECAR), which represents glycolytic pathway activity, were increased during BA differentiation (Fig 1G and H). However, because the rate of increase in basal OCR was higher than that of ECAR, the basal OCR/ECAR ratio was significantly higher in BAs than in brown preadipocytes (Fig 1I). These observations suggest that ATP production is dominantly mediated by the OXPHOS pathway in BAs. Consistent with these findings, the expression of OXPHOS complex members (respiratory chain complexes I, II, III, IV, and V) was clearly higher in BAs than in brown preadipocytes (Fig 1J). Moreover, not only the OXPHOS complexes, including cytochrome c oxidase subunit IV (COX4) but also other mitochondrial proteins, including UCP1, cytochrome c (Cyt C), and translocase of the outer membrane (Tom) 20, were markedly increased during differentiation (Fig 1K). Conversely, the amounts of the translocon component Sec61α and the ER chaperone GRP78 were reduced (Fig 1K). Given all these findings, we conclude that BAs acquire developed mitochondria to enable ATP production mainly via activation of the OXPHOS pathway. Interestingly, the thinned ER was surrounded by mitochondria (Fig 1B, day 6). To quantify this finding, we measured contact sites, which were defined as sites at which the distance between the membranes of the two organelles was less than 30 nm (Fig 1B, yellow lines). The areas of ER–mitochondria contact sites, which were hardly detectable in brown preadipocytes (Fig 1B, day 0), were significantly increased during BA differentiation (Fig 1L). The ratios of ER–mitochondria contact sites to both the ER perimeter and the mitochondrial perimeter were significantly increased (Fig 1M and N), suggesting that the ER and mitochondrial membranes may actively contact each other rather than meeting simply because of mitochondrial expansion. We, thus, hypothesize that ER–mitochondria crosstalk may play a role in the function of BAs.

## Phosphorylation of PERK during BA differentiation

We next examined the involvement of the UPR signalling pathway in BAs. The amounts of PERK and IRE1α decreased as BA differentiation proceeded because of the reduction in ER area (Fig S2). However, detailed time course experiments revealed that PERK band shifts were temporarily present after 6–12 h of culture with differentiation enhancement medium on day 2 (Fig 2A, top panel). These band shifts were eliminated by treatment with protein phosphatase (Fig 2B and C). Autophosphorylation of PERK was also detected with an antibody against phosphorylated PERK at Thr980 (Fig 2A, second panel). These findings suggest that the induction of BA differentiation triggers temporal phosphorylation, including autophosphorylation, of PERK on day 2. However, the electrophoresis pattern of IRE1α was not affected during differentiation, and expression of the spliced form of XBP1, which is the result of activated IRE1α, was not observed (Fig 2A and B). Because an antibody that specifically recognizes mouse ATF6 is not available, we examined the induction of GRP78, a downstream target of ATF6, and observed little changes in its expression during differentiation (Fig 2B). Our findings suggest that PERK is specifically phosphorylated among UPR sensors during BA differentiation. Under ER stress conditions, activated PERK phosphorylates eIF2α, resulting in attenuation of global protein translation to reduce the ER protein load while promoting the specific translation of ATF4 (Harding et al,

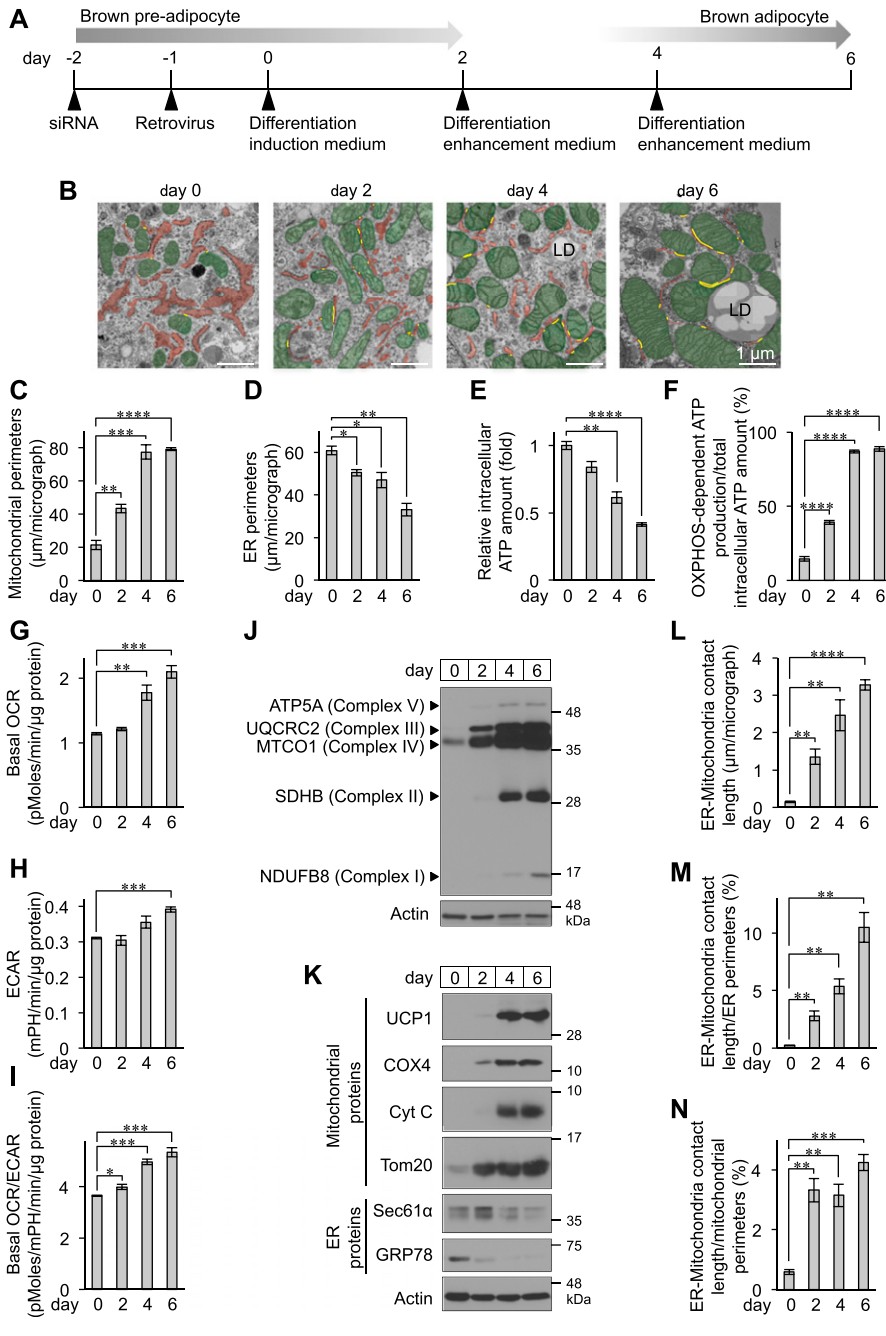

**Figure 1. Morphological changes in the ER and mitochondria during brown adipocyte (BA) differentiation.**

**(A)** Schematic representation of the BA preparation method used in this study. Primary brown preadipocytes were grown to confluence (day 2) and cultured with differentiation induction medium beginning on day 0. After initiation of differentiation, the cell culture medium was changed to differentiation enhancement medium on day 2 and replaced with fresh medium on day 4. Cells on day 6 were used as differentiated BAs. siRNA transfection and retroviral transduction were performed on the indicated days. **(B)** Electron micrographs (EMs) of brown preadipocytes and BAs on day 0, 2, 4, and 6. Mitochondria and the ER have been false-coloured in green and red, respectively. The yellow lines denote ER–mitochondria contact sites (sites with <30 nm of distance between two membranes). **(C, D)** Quantification of the ER (C) and mitochondrial (D) perimeters in EMs. Total length of the ER or mitochondrial perimeters in one EM was calculated with ImageJ software. Data are shown as the average from over 20 EMs (*n* = 3 independent experiments). **(E, F)** Quantification of intracellular ATP content in BAs during differentiation. The cells were treated with or without 1 *μ*g/ml oligomycin A for 45 min. **(E, F)** OXPHOS-dependent ATP production (F) was calculated from the total intracellular ATP content (E) and the oligomycin A–insensitive intracellular (glycolysis-dependent) ATP content. The data are shown as the percentage relative to total intracellular ATP content at each day (*n* = 3 independent experiments). **(G, H, I)** Measurement of basal oxygen consumption rate (OCR) (G) and extracellular acidification rate (ECAR) (H). **(I)** On day 0, 2, 4, and 6, the OCR and ECAR of brown preadipocytes and BAs were measured using a Seahorse analyser and XF Cell Mito stress test kit, and the basal OCR/ECAR (I) were calculated as described in the Materials and Methods section. Data were normalized by total protein content (*n* = 3 independent experiments). **(J, K)** Expression levels of OXPHOS complexes (J) and mitochondrial and ER proteins (K) during BA differentiation. The cells were lysed on the indicated days and analysed by immunoblotting using the indicated antibodies. GPR94 and GRP78 protein was detected by a KDEL antibody. Actin was used as a loading control. **(L, M, N)** Quantification of ER–mitochondria contact site in EMs. **(L)** Total length of contact site in one EM was calculated with ImageJ software (L). The data are shown as the average from 20 EMs (*n* = 3 independent experiments). **(M, N)** ER–mitochondria contact length was normalized to the ER (M) or mitochondrial (N) perimeter and shown as percentage. Data information: data are presented as mean ± SEM. *P < 0.05, **P < 0.01, ***P < 0.001, ****P < 0.0001 (*t* test). LD, lipid droplet.

1999; Vattem & Wek, 2004). However, the levels of newly synthesized polypeptides, which were labeled with puromycin, were not reduced in differentiating cells (Fig 2D). The expression of the ATF4 protein and its target genes, *CHOP* and *Gadd34*, was also not increased (Fig 2E–G). Collectively, our results suggest that PERK may play a role via a pathway other than the eIF2*α*–ATF4 axis in BA differentiation.

### Requirement of PERK for mitochondrial development in BAs

To investigate the role of PERK in BAs, we first assessed the requirement of PERK for adipogenesis. The total amounts of LDs in PERK-deficient cells were comparable with those in control cells (Fig 3A and B). Multiple LDs were observed not only in control siRNA-transfected (sictrl) cells but also in PERK-deficient BAs by electron microscopy (Fig 3C). Confocal microscopy revealed that the total areas of the ER and mitochondria in BAs were not affected by PERK deficiency (Fig S3A–C). Consistent with the results for the mitochondrial area, the amount of mitochondrial DNA relative to that of nuclear DNA did not differ between PERK-deficient BAs and control BAs (Fig S3D). Measurement of the organellar perimeter by electron microscopy revealed that the total ER perimeter was not affected by PERK deficiency, and a marginal but nonsignificant decrease in

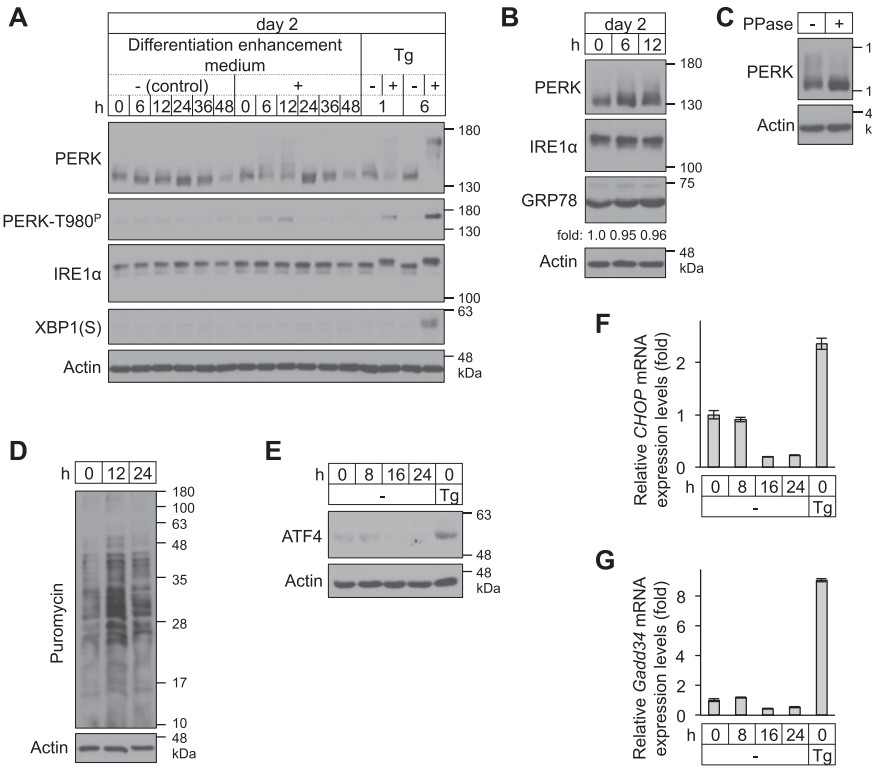

**Figure 2. Phosphorylation of PERK during brown adipocyte differentiation.**
**(A, B, C)** Phosphorylation of PERK in differentiating cells. **(A, B)** Cells were cultured with differentiation medium (A, − [control]) or differentiation enhancement medium (A, + and B) for the indicated number of hours on day 2. **(A)** Cells treated with (+) or without (−) 40 nM thapsigargin (Tg) for 1 or 6 h on day 2 were included as positive controls (A). **(A, B)** The cell lysates were analysed by immunoblotting (IB) with the indicated antibodies (A, B). **(C)** Lysates from cells cultured with differentiation enhancement medium for 12 h on day 2 were incubated with or without 2 units of λ phosphatase (PPase) at 30°C for 30 min and analysed by IB with the indicated antibodies (C). **(B)** The expression of GRP78 was calculated and is shown as the ratio relative to actin expression (B). **(D)** Newly synthesized proteins in differentiating cells. Cells were incubated with 10 μg/ml puromycin for 10 min and lysed at the indicated time points on day 2. Newly synthesized proteins were detected by IB with an anti-puromycin antibody. **(E)** IB analysis of ATF4 in differentiating cells. Cells were harvested after the indicated number of hours on day 2 and analysed by IB using the indicated antibodies. Cells treated with 40 nM Tg for 6 h on day 2 were included as positive controls. **(F, G)** Expression of *CHOP* (F) or *Gadd34* (G) mRNA in differentiating cells. Cells were harvested after the indicated number of hours on day 2, and total RNA was isolated. The data are shown as the fold change relative to the value at 0 h (n = 3 independent experiments). Cells treated with 40 nM Tg for 12 h on day 2 were included as positive controls brown adipocytes.

mitochondrial perimeter was observed in PERK-deficient BAs (Fig S3E–G). The areas of the ER–mitochondria contact sites and the ratios of the contact sites to the ER or mitochondrial perimeters in PERK-deficient BAs were comparable with those in control BAs (Fig S3H–J). Interestingly, electron microscopy revealed marked differences in the morphology of mitochondrial cristae among BAs (Fig 3D). The mitochondria in the control cells possessed dense parallel cristae, consistent with a previous report (Napolitano & Fawcett, 1958), whereas significantly fewer mitochondria possessed dense parallel cristae in PERK-deficient BAs (Figs 3E and S3K). We next examined the expression levels of mitochondrial and ER proteins by immunoblotting (IB). Marginal increases in UPR-related ER protein levels were observed in PERK-deficient cells compared with control cells on day 0 (Fig 3F). The slight reduction in the outer membrane protein Tom20 expression on day 6 may have depended on the marginal reduction in mitochondrial perimeter (Figs 3F and S3G). Drastic changes were also observed: the expression levels of mitochondrial inner membrane proteins (UCP1 and COX4) and a crista protein (Cyt C) were markedly reduced in PERK-deficient BAs (Fig 3F). Taken together, our findings suggest that PERK is required for crista formation and the expression of inner membrane and crista proteins during BA differentiation.

### Requirement of PERK for OXPHOS-dependent ATP production in BAs

Next, to examine whether PERK deficiency affects mitochondrial functions, BAs were stained with a membrane potential–indicating

dye (Fig 4A) and ΔΨm was quantified using a fluorescence-activated cell sorter (Fig 4B). The number of BAs exhibiting low ΔΨm was significantly increased by PERK deficiency (Fig 4C). The total amount of intracellular ATP and the ratio of OXPHOS-dependent ATP production were reduced by PERK deficiency in BAs (Fig 4D and E). Moreover, the basal OCR, but not the ECAR, was significantly reduced in PERK-deficient BAs, resulting in a reduction in the basal OCR/ECAR ratio (Fig 4F–H). These phenotypes were attributable to the defective crista structure and the reduced expression of COX4 and Cyt C in PERK-deficient BAs (Fig 3E and F). Collectively, these results strongly suggest that PERK contributes to OXPHOS-dependent ATP production via regulation of crista formation and mitochondrial inner membrane biogenesis in BAs.

### Requirement of PERK phosphorylation for mitochondrial functions, but not for the UPR, in BAs

In response to ER stress, activation of PERK is triggered by dissociation of GRP78 from its ER luminal domain, after which PERK is oligomerized and autophosphorylated (Bertolotti et al, 2000). To examine whether the phosphorylation of PERK during BA differentiation is caused by the ER stress, we created a retrovirus-encoded PERK mutant with deletion of the ER luminal domain (PERK-ΔLD), which does not respond to ER stress and used it to infect PERK-deficient BAs (Figs 5A and S4A). The deletion of the ER luminal domain had no effect on the phosphorylation of PERK (Fig 5B, top panel, parentheses). In addition, a band shift of a catalytically inactive PERK mutant (PERK-ΔLD-KA) that represents the

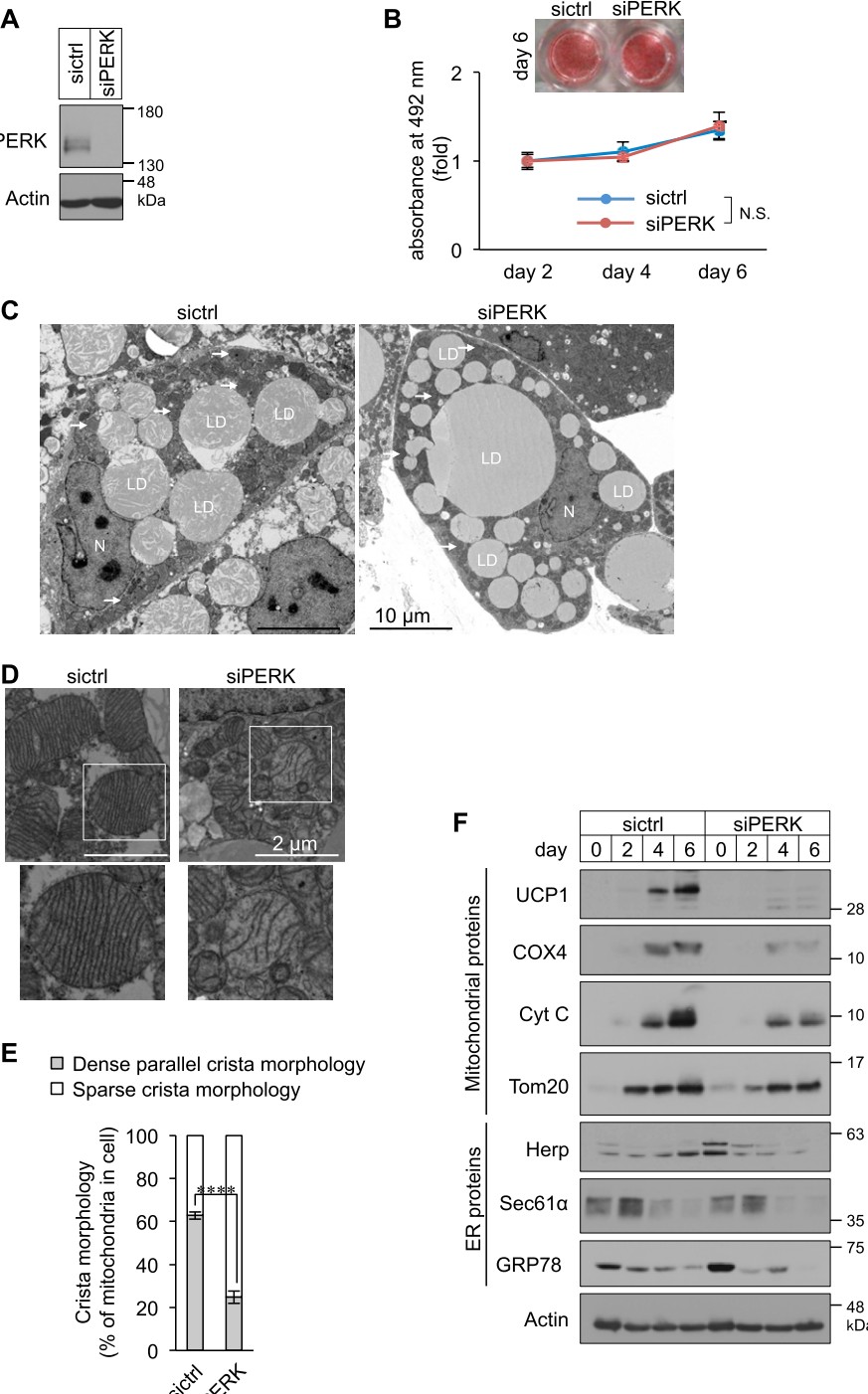

**Figure 3. Requirement of PERK for mitochondrial development in brown adipocytes (BAs).**
**(A)** Knockdown efficiency of PERK in BAs. siRNA-transfected cells were lysed on day 6 and analysed by immunoblotting with an anti-PERK antibody. **(B)** PERK-independent adipogenesis. siRNA-transfected BAs were stained with oil red O (images), and the isolated oil red O from cells was measured by quantification of light absorbance at 492 nm (graph). Data are shown as the fold change relative to the value on day 2 (graph) (n = 3 independent experiments). **(C, D, E)** Requirement of PERK for dense parallel cristae formation in mitochondria. **(C, D, E)** Brown preadipocytes were transfected with sictrl or siPERK, fixed on day 6, and analysed by electron microscopy (arrows, mitochondrion; LD, lipid droplet; N, nucleus) (C, D, E). Mitochondria were divided into mitochondria possessing dense parallel cristae and mitochondria possessing sparse cristae and were counted (in a total of 50 individual cells) (E). The white square denotes the magnified region (n = 3 independent experiments). **(F)** Requirement of PERK for mitochondrial (inner membrane and crista) and ER proteins. Brown preadipocytes were transfected with siRNAs. The lysates were analysed by immunoblotting with the indicated antibodies. GPR94 and GRP78 protein was detected by a KDEL antibody. **(B, E)** Data information: data are presented as mean ± SEM. NS; ****$P < 0.0001$ (repeated measures ANOVA in (B), $t$ test in (E)).

phosphorylated form was observed in differentiating PERK-deficient cells (Fig 5C). Considering the observation of PERK autophosphorylation (Fig 2A), our findings suggest that PERK is phosphorylated, presumably by a non-autophosphorylation mechanism in addition to autophosphorylation, during BA differentiation independently of the ER stress. We next investigated the phosphorylated amino acid residues of PERK in differentiating cells. Liquid chromatography coupled with tandem mass spectrometry (LC-MS/MS)–based phosphoproteomic analysis was performed using immunopurified Flag-tagged PERK-ΔLD-KA. Ser719 in the kinase insert region was revealed to be a strong candidate amino acid residue for phosphorylation during BA differentiation (Figs S4B and 5A). Database analysis using Phospho-SitePlus (https://www.phosphosite.org/homeAction) suggested that Ser715 might also be phosphorylated. Because the set of Ser715, Ser717

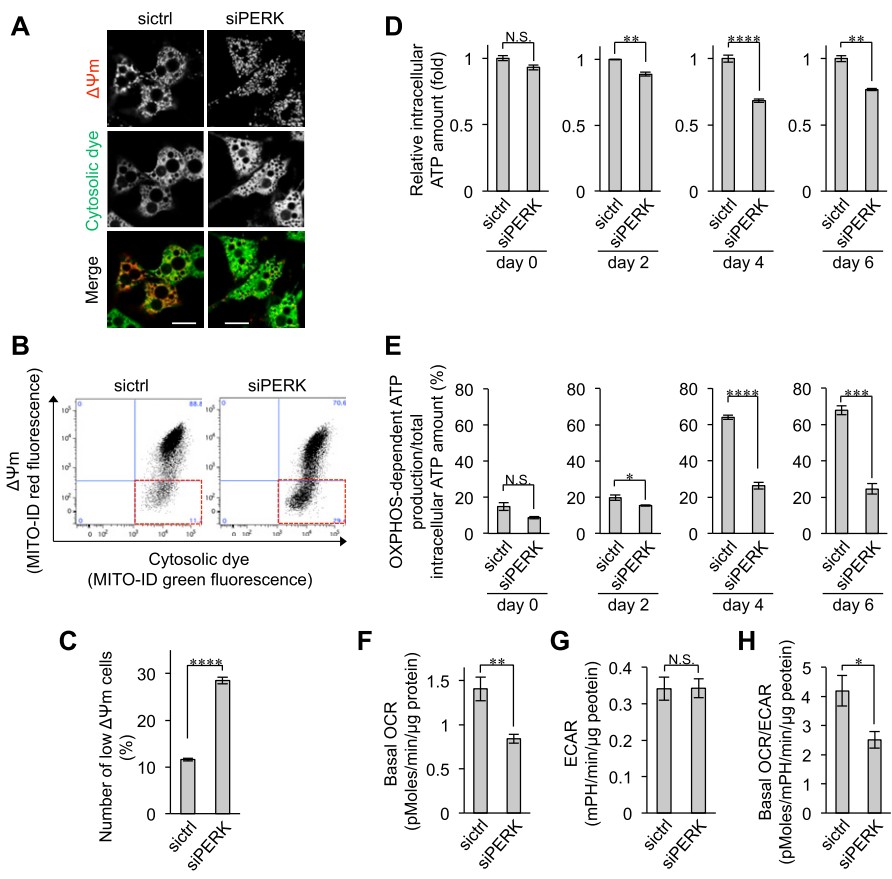

**Figure 4. Requirement of PERK for ΔΨm and OXPHOS-dependent ATP production in brown adipocytes (BAs).**

**(A)** Fluorescence images of ΔΨm in BAs. siRNA-transfected BAs were treated with MITO-ID for 15 min at room temperature on day 6. The ΔΨm was visualized by confocal fluorescence microscopy. The intensity of red fluorescence denotes the ΔΨm. Green fluorescence denotes cytosolic dye, which was used as the loading control. Scale bar, 5 $\mu$m. **(B, C)** Requirement of PERK for maintenance of ΔΨm. siRNA-transfected BAs were stained with MITO-ID and necrosis detection reagents for 15 min at room temperature on day 6. **(B)** Necrotic cells were removed, and the population of cells showing low ΔΨm was analysed by flow cytometry (B, more than 10,000 cells for each sample). **(C)** The cell population with low ΔΨm was surrounded with the dotted red square and counted (C). Data are shown as the percentages relative to the total number of cells ($n$ = 3 independent experiments). **(D, E)** Requirement of PERK for OXPHOS-dependent ATP production. **(D, E)** sictrl- or siPERK-transfected cells were harvested on the indicated days, and the total intracellular ATP content (D) and OXPHOS-dependent ATP production (E) were measured as described in Fig 1E and F ($n$ = 3 independent experiments). **(F, G, H)** Requirement of PERK for basal oxygen consumption rate (OCR) in BAs. The OCR and extracellular acidification rate of sictrl- or siPERK-transfected cells were measured using a Seahorse analyser and XF Cell Mito stress test kit. **(F, G, H)** Data were normalized by total protein content and show the basal OCR (F), extracellular acidification rate (G), and basal OCR/ECRA (H) ($n$ = 3 independent experiments). Data information: data are presented as mean ± SEM. NS; *$P < 0.05$; **$P < 0.01$; ***$P < 0.001$; ****$P < 0.0001$ ($t$ test).

and Ser719 (3S) is conserved in rodents and humans (Fig 5D), a polyclonal antibody against a peptide containing phosphorylated 3S (PSPERS[P]RS[P]FS[P]VGI) was raised in rabbits and was designated as an antibody against phosphorylated PERK at Ser715, Ser717, and/or Ser719 (PERK-3S[P] Ab). PERK-3S[P] Ab successfully detected endogenous PERK and exogenous PERK-ΔLD, but not PERK-ΔLD-3SA (in which serine was substituted with alanine at Ser715, Ser717, and Ser719), during BA differentiation (Fig 5B, second panel). Our results using PERK-3S[P] Ab provide evidence that PERK is physiologically phosphorylated in the kinase insert region, at least at Ser719, during differentiation.

We next investigated the requirements of Ser715, Ser717, and Ser719 for mitochondrial function. The reduced expression of UCP1, COX4, and Cyt C and the reduced ΔΨm and OXPHOS-dependent ATP production caused by PERK deficiency were ameliorated in cells transfected with PERK-ΔLD but not in those transfected with PERK-ΔLD-KA or PERK-ΔLD-3SA (Fig 5E–G), suggesting that both kinase activity and phosphorylation at Ser715, Ser717, and/or Ser719 may be required for PERK-mediated mitochondrial inner membrane protein biogenesis and functions in BAs. Although PERK was also recognized by PERK-3S[P] Ab in thapsigargin- or tunicamycin-treated cells (Fig S4C), attenuation of ER stress-induced transcriptional activation by ATF4, which was measured using an amino acid response element-luciferase reporter (AARE-luc) (Miyake et al, 2016), in PERK-deficient BAs was completely eliminated by exogenously expressed PERK-3SA, as it was by wild-type (WT) PERK (Fig 5H). Collectively, our observations suggest that the kinase activity and

Ser715, Ser717, and/or Ser719 residues of PERK are indispensable for mitochondrial functions but not for the UPR.

## Involvement of GABPα in PERK-mediated mitochondrial inner membrane biogenesis

To examine whether PERK regulates the transcription of BA-related genes, total RNA harvested from control and PERK-deficient BAs on day 4 was analysed using quantitative reverse transcriptase-PCR (qPCR). PERK deficiency significantly reduced the mRNA levels of *Cox4i1*, *Cox8b*, *Cox7a1*, and *Cycs*, which are the genes associated with OXPHOS complexes, and *Ucp1*, *Cidea*, and *Dio2*, which are the genes related to BA function (Fig 6A). PERK was also required for the expression of *Ppargc1a* mRNA, which encodes the transcriptional coactivator peroxisome proliferator-activated receptor γ coactivator-1α (PGC-1α) that regulates OXPHOS complex genes and *Ucp1*, but not for the expression of *Ppargc1b* mRNA (Fig 6A). In contrast, the mRNA expression of *Pparg* and *Prdm16*, which are the master regulator genes of BA differentiation, was not changed by PERK deficiency (Fig 6A). These results suggest that PERK specifically regulates the transcriptional expression of subsets of mitochondria-related genes. To understand the mechanism by which PERK regulates gene expression, we performed RNA sequencing analysis of control and PERK-deficient cells after 16 or 24 h of culture with differentiation enhancement medium on day 2. PERK deficiency reduced the expression levels of 381 genes at both time points (by >1.32-fold) (Fig 6B and Table S1). We next analysed the conserved transcription

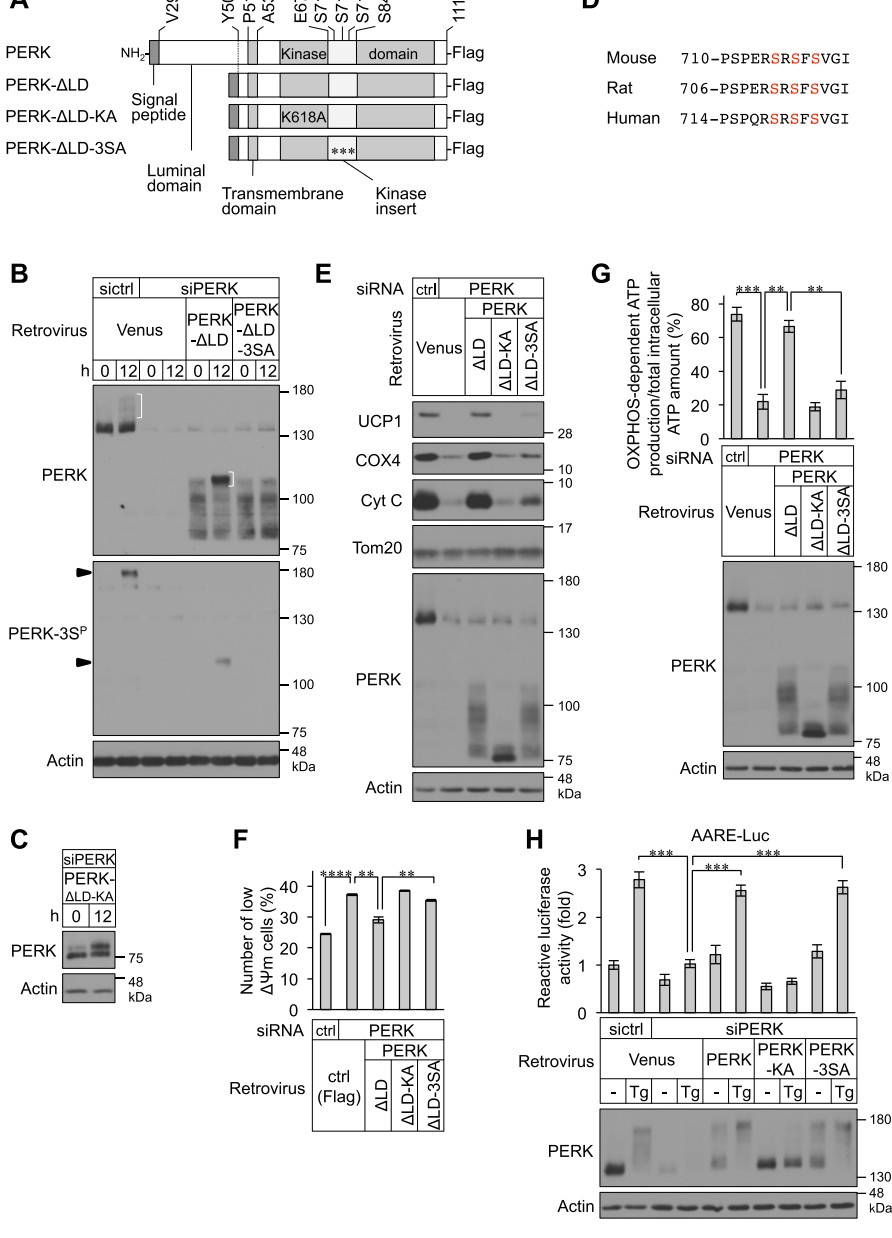

**Figure 5. Requirement of PERK-3S phosphorylation for mitochondrial functions, but not for the unfolded protein response, in brown adipocytes (BAs).**

**(A)** Schematic representation of the C-terminal Flag-tagged full-length PERK, luminal domain-deleted (ΔLD; ΔVal29-Tyr502) mutant PERK, kinase-negative ΔLD (ΔLD-KA; substitution of lysine to alanine at Lys618) mutant PERK, and 3SA ΔLD (ΔLD-3SA; substitution of serine to alanine at Ser715, Ser717, and Ser719) mutant PERK. **(B)** Immunoblotting (IB) analysis of the phosphorylation of endogenous or exogenously expressed PERK in differentiating cells. siRNA-transfected cells were infected with retroviruses expressing Venus, PERK-ΔLD, or PERK-ΔLD-3SA and cultured with differentiation enhancement medium for the indicated number of hours on day 2. The cell lysates were analysed by IB with anti-PERK and anti-PERK-3S$^P$ antibodies. Actin was used as a loading control. Arrowheads indicate the phosphorylated PERK at Ser715, Ser717, and/or Ser719. White parentheses indicate the phosphorylated PERK during BA differentiation. **(C)** IB analysis of exogenously expressed PERK-ΔLD-KA in differentiating cells. siPERK-transfected cells were infected with retroviruses expressing PERK-ΔLD-KA. The cells were lysed after the indicated number of hours on day 2 and analysed by IB with the indicated antibodies. **(D)** Alignment of the backbone peptide of PERK for the generation of phospho-Ser715, phospho-Ser717, and phospho-Ser719 antibodies from the indicated various species. Amino acids conserved with Ser715, Ser717, and Ser719 in mouse PERK are highlighted in red. **(E)** Requirement of PERK kinase activity and phosphorylation at Ser715, Ser717, and/or Ser719 for the expression of mitochondrial inner membrane and crista proteins. siRNA-transfected cells were infected with retroviruses expressing Venus, PERK-ΔLD, PERK-ΔLD-KA, or PERK-ΔLD-3SA, lysed on day 6 and analysed by IB with the indicated antibodies. **(F)** Requirement of PERK kinase activity and phosphorylation at Ser715, Ser717, and/or Ser719 for maintenance of ΔΨm in BAs. siRNA-transfected primary brown preadipocytes were infected with the indicated retroviruses. The ΔΨm values were analysed by flow cytometry (more than 10,000 cells for each sample) and are shown as described in Fig 4C (n = 3 independent experiments). **(G)** Requirement of PERK kinase activity and phosphorylation at Ser715, Ser717, and/or Ser719 for OXPHOS-dependent ATP production. siRNA-transfected primary brown preadipocytes were infected with the indicated retroviruses. The ATP levels were measured (top) and are shown as described in Fig 1F. The cell lysates were analysed by IB with the indicated antibodies (bottom) (n = 3 independent experiments). **(H)** Role of PERK phosphorylation at Ser715, Ser717, and/or Ser719 on the eIF2α-ATF4 pathway in differentiating cells. siRNA-transfected brown preadipocytes were co-transfected with 10 μg of AARE-luc and 1 μg of Renilla-luc and infected with the indicated retroviruses expressing Venus, PERK, PERK-KA, or PERK-3SA. The cells were stimulated with 40 nM Tg for 12 h on day 2, and the relative luc activity was measured (top). AARE-luc activity was normalized to Renilla-luc activity. Data are shown as the fold change relative to the value in non-stimulated sictrl-transfected BAs. The cell lysates were analysed by IB with the indicated antibodies (bottom) (n = 3 independent experiments). Data information: data are presented as mean ± SEM. **P < 0.01, ***P < 0.001, ****P < 0.0001 (t test).

factor–binding motifs of these genomic DNA sequences. The binding motifs of transcription factor EB, cAMP response element–binding protein (CREB) and GABPα were enriched in 381 genes regulated by PERK in differentiating cells (Fig 6C). Similar to the case in differentiated BAs, phosphorylated PERK was detected by PERK-3S$^P$ Ab in HEK293 cells habituated to OXPHOS through culture with no-glucose medium (Fig S5A and B). RNA sequencing analysis of HEK293 cells revealed that the expression levels of 630 genes were lower in PERK-KO cells than in WT cells (by >1.49-fold) (Table S2). Among them, 111 genes were categorized as mitochondria-related genes by gene

annotation enrichment analysis (Fig S5C), and the binding motif of GABPα was found to be enriched in 48 mitochondria-related genes in HEK293 cells (Fig S5D). GABPα regulates the transcription of OXPHOS complex genes, including Cycs and Cox4i1, and functions as a regulator of glycolytic beige adipocyte differentiation (Chen et al, 2009, 2019). Taken together, the results obtained from BAs and HEK293 cells as well as those from previous reports suggest that PERK may activate the GABPα pathway, resulting in mitochondrial inner membrane biogenesis in BAs. To investigate this hypothesis, we examined whether PERK is required for transcriptional activation by GABPα in

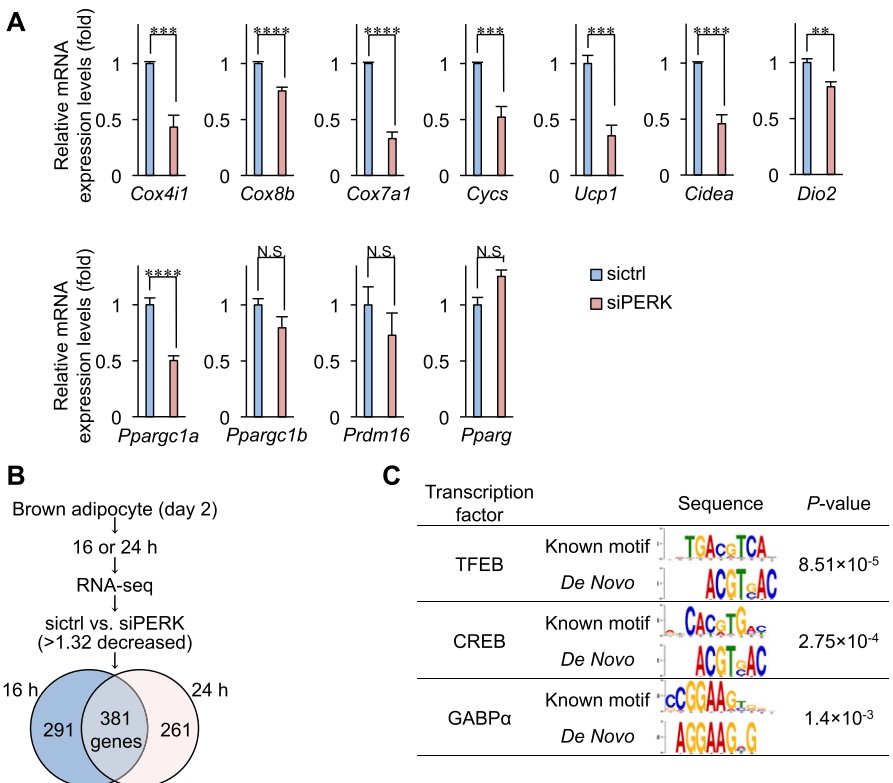

**Figure 6. Mitochondrial gene expression in PERK-deficient brown adipocytes.**
**(A)** qPCR analysis of brown adipocyte-related genes in differentiating cells. Total RNA from siRNA-transfected cells was isolated on day 4. The gene expression levels were analysed by qPCR. The mRNA expression of the indicated genes was normalized to that of *S18*. Data are shown as the fold change relative to the value in sictrl-transfected cells (*n* = 3 independent experiments). **(B)** Venn diagram of common genes between two time points (sictrl versus siPERK, >1.32-fold decrease). sictrl- or siPERK-transfected differentiating cells were cultured with differentiation enhancement medium for 16 or 24 h, and mRNA was isolated. RNA sequencing data were analysed with a Venn diagram. **(C)** Motif enrichment analysis of PERK-related genes in differentiating cells. Data information: data are presented as mean ± SEM. NS; \*\**P* < 0.01; \*\*\**P* < 0.001; \*\*\*\**P* < 0.0001 (*t* test).

differentiating cells with a luciferase assay using the promoter region of mitochondrial translation initiation factor 2 containing the binding site for GABPα (Hayashi et al, 2007). The expression level of GABPα protein was not affected by PERK deficiency (Fig S5E); however, transcriptional activation by GABPα was significantly reduced in PERK-deficient BAs, whereas it was not reduced in IRE1α- or ATF6-deficient cells (Fig 7A and B). This effect was attenuated by exogenously expressed PERK-ΔLD but not PERK-ΔLD-KA or PERK-ΔLD-3SA (Fig 7C), suggesting that PERK specifically regulates the GABPα pathway as a result of PERK kinase activation and phosphorylation at Ser715, Ser717, and/or Ser719 during BA differentiation. We next examined whether the PERK–GABPα axis regulates mitochondrial inner membrane protein biogenesis and function in BAs. The reduced expression of UCP1, COX4, and Cyt C and inhibition of OXPHOS-dependent ATP production in PERK-deficient BAs were ameliorated by coexpression of GABPα with GA-binding protein subunit *β*-1 (GABPβ1), which is a coactivator of GABPα (Fig 7D, lanes 2 and 4). UCP1, COX4, and Cyt C expression and OXPHOS-dependent ATP production were reduced by GABPα deficiency just as they were by PERK deficiency (Fig S5F). PERK-ΔLD did not reconstitute the phenotype in GABPα-deficient BAs (Fig 7D, lanes 5 and 6), suggesting that GABPα functions downstream of PERK in mitochondrial inner membrane biogenesis and function.

### Requirement of PERK for thermogenesis of BAs

One of the most important roles of mitochondria in BAs is thermogenesis. To clarify the role of PERK in intracellular thermogenesis,

we performed a thermogenic analysis using a fluorescent polymeric thermometer in single BAs (Gota et al, 2009). Treatment of differentiated BAs with the *β*₃-adrenoceptor (*β*₃AR) agonist CL316,243 increased the intracellular temperature in control BAs but not in PERK-deficient BAs (Fig 8A). The percentage of retrovirus-infected BAs expressing the GFP-variant Venus was ~67% (data not shown). At this infection efficiency, the thermogenic defect in PERK-deficient BAs was significantly improved by exogenously expressed PERK-ΔLD (Fig 8A). However, PERK-ΔLD-3SA had no effect on the phenotype (Fig 8A). Considering the result that PERK-ΔLD-KA did not rescue the reduced expression of UCP1 in PERK-deficient BAs at all (Fig 5E), PERK kinase activation and phosphorylation at Ser715, Ser717, and/or Ser719 may be indispensable for thermogenesis.

Finally, we examined the requirement of PERK for thermogenesis in vivo. In newborn mice, iBAT thermogenesis is essential for body temperature maintenance (Liu et al, 2003). We analysed iBAT derived from newborn mice within 12 h after birth to avoid the effect of hyperglycaemia due to progressive degeneration of pancreatic secretory cells in *PERK*⁻/⁻ mice (Zhang et al, 2002). Although there were no obvious differences in gross and histological observations (Fig S6A and B), electron microscopic analysis revealed that the numbers of mitochondria with dense parallel cristae were significantly decreased in iBAT derived from *PERK*⁻/⁻ mice compared with that derived from WT mice (Fig 8B). In accord with the results of the in vitro experiments, the expression levels of UCP1, COX4, and Cyt C, but not Tom20, were significantly reduced in iBAT derived from *PERK*⁻/⁻ mice (Fig 8C–G). The temperature of the skin overlying iBAT in newborn mice reflects the extent of iBAT thermogenesis (Hodges

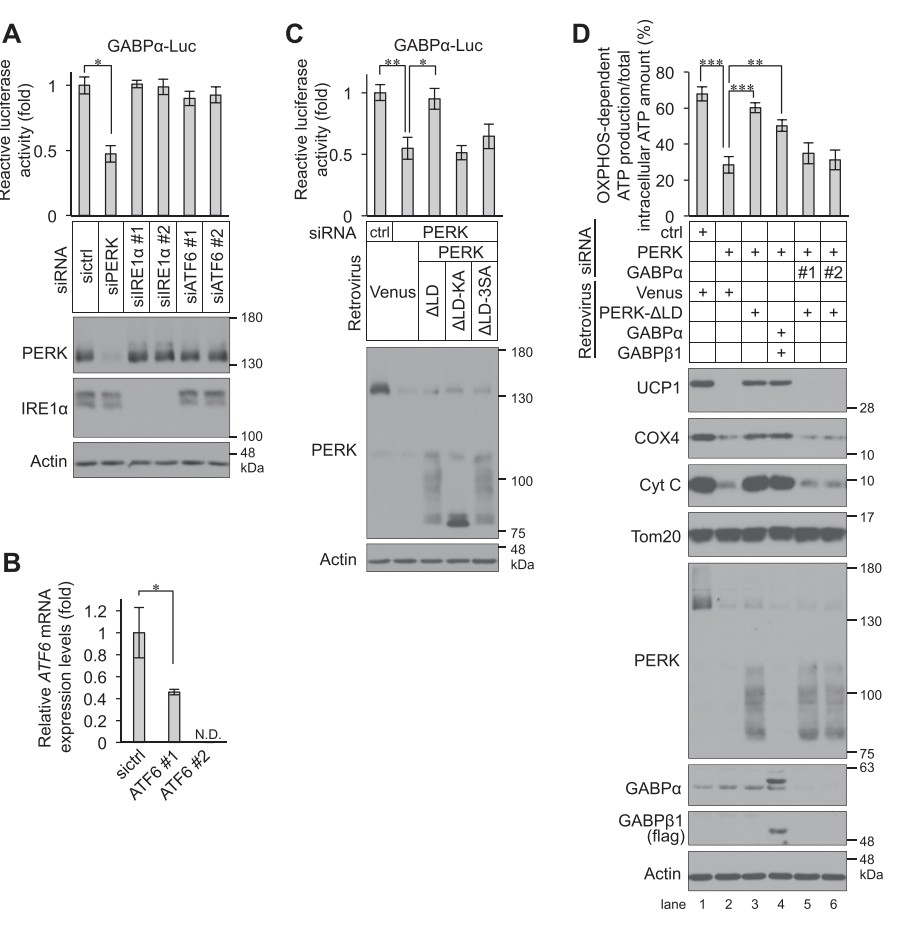

**Figure 7. Involvement of GABPα in PERK-mediated mitochondrial inner membrane protein biogenesis.**
**(A)** Role of inositol-requiring enzyme 1α or ATF6 in the transcriptional activity of GABPα in differentiating cells. siRNA-transfected brown preadipocytes were co-transfected with GABPα-luc and Renilla-luc. Relative luc activity was measured on day 3 (top). GABPα-luc activity was normalized to Renilla-luc activity. Data are shown as the fold change relative to the value in sictrl-transfected brown adipocytes (BAs) (n = 3 independent experiments). The knockdown efficiency of siPERK and siRNAs against inositol-requiring enzyme 1α (siIRE1α #1 and siIRE1α #2) was analysed by immunoblotting (IB) (bottom). **(B)** Knockdown efficiency of ATF6 siRNA (siATF6) transfection. Primary brown preadipocytes were transfected with sictrl or with ATF6 #1 or ATF6 #2 siRNA. The amount of *ATF6* mRNA on day 3 was analysed by qPCR and normalized to that of *S18* mRNA. The data are shown as the fold change relative to the value in sictrl-transfected BAs (n = 3 independent experiments). **(C)** Requirement of PERK for the transcriptional activity of GABPα in differentiating cells. siRNA-transfected brown preadipocytes were co-transfected with 10 μg of GABPα-luc and 1 μg of Renilla-luc and infected with retroviruses expressing Venus, PERK-ΔLD, PERK-ΔLD-KA, or PERK-ΔLD-3SA. The relative luc activity was measured on day 3 (top). GABPα-luc activity was normalized to Renilla-luc activity. Data are shown as the fold change relative to the value in sictrl-transfected BAs (n = 3 independent experiments). The cell lysates were analysed by IB with the indicated antibodies (bottom). **(D)** Requirement of the PERK–GABPα pathway for the expression of mitochondrial proteins and OXPHOS-dependent ATP production. ATP levels were measured (top) and are shown as described in Fig 1F. Data are shown as the percentage relative to total intracellular ATP content in each transfected BA (n = 4 independent experiments). siRNA-transfected cells were infected with the indicated retroviruses, lysed on day 6, and analysed by IB with the indicated antibodies (bottom). Data information: data are presented as mean ± SEM. *P < 0.05, **P < 0.01, ***P < 0.001 (t test). N.D., not detectable.

et al, 2008). iBAT thermogenesis was, thus, measured by using an infrared thermographic imaging system. During exposure to 16°C, newborn mice experienced cold stress-induced hypothermia, which was exacerbated in *PERK*[−/−] mice (P = 0.01037) (Fig 8H). Collectively, our observations strongly suggest that body temperature is regulated by BAT-derived thermogenesis through PERK-mediated mitochondrial inner membrane protein biogenesis in mice exposed to cold stress.

# Discussion

In this work, we provide evidence that the ER-resident kinase PERK is essential for mitochondrial development in BAs. PERK is phosphorylated at Ser715, Ser717, and/or Ser719 by a mechanism that is independent of ER stress during BA differentiation. PERK kinase activity and its phosphorylation at Ser715, Ser717, and/or Ser719 are required for the expression of mitochondrial inner membrane and crista proteins, including the OXPHOS complexes and UCP1, as a result of transcriptional activation by GABPα. We also propose a novel role in which PERK aids in OXPHOS-dependent ATP production and UCP1-mediated thermogenesis in BAs. PERK has been reported to regulate mitochondrial functions and homeostasis

through the conventional PERK–eIF2α–ATF4 axis in cells other than BAs (Hori et al, 2002; Bouman et al, 2011; Rainbolt et al, 2013; Lebeau et al, 2018). A recent study has shown that the PERK–ATF4 axis increases the expression of supercomplex assembly factor 1, which promotes the formation of mitochondrial respiratory supercomplexes under ER and nutrient stress conditions (Balsa et al, 2019). However, increased expression of ATF4 and induction of downstream target genes of ATF4 were not observed on day 2 (Fig 2E–G). Thus, we conclude that mitochondrial inner membrane protein biogenesis in BAs is regulated by a novel PERK–GABPα axis. However, several questions regarding the upstream and downstream targets of this mechanism must be answered by further investigations. One important uncertainty is the location at which PERK recognizes mitochondrial conditions during BA differentiation. To address this uncertainty, it will be necessary to visualize the sites at which PERK is phosphorylated by the unidentified kinase. The polyclonal PERK-3S[P] Ab used in this manuscript is useful for the detection of phosphorylated PERK by IB but not by immunocytochemistry. Successful generation of another PERK-3S[P] Ab that is useful for immunocytochemistry and/or immunoelectron microscopy would enable investigation of this issue. The mediator that conveys the alterations in mitochondrial conditions to PERK or its upstream kinase also remains to be identified. In BAs treated with the

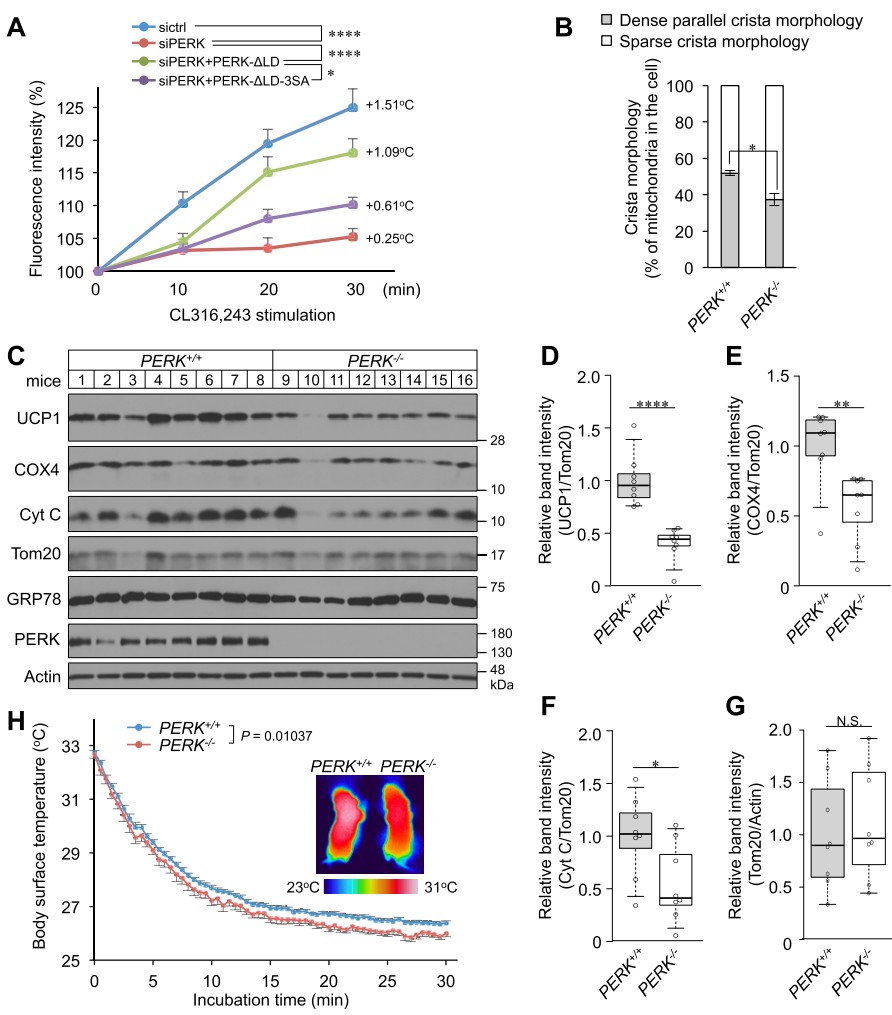

**Figure 8. Requirement of PERK for thermogenesis.**
**(A)** Requirement of PERK for $\beta_3$AR stimulation-induced intracellular thermogenesis. siRNA-transfected brown preadipocytes were infected with retroviruses expressing PERK-ΔLD or PERK-ΔLD-3SA, incubated with cellular thermoprobes on day 6, stimulated with 0.5 $\mu$M CL316,243 and observed by confocal fluorescence microscopy. The intracellular temperature was analysed by ImageJ software. Data are shown as the fold change relative to the value at 0 min (sictrl, 20 individual cells; siPERK, 19 individual cells; siPERK+PERK-ΔLD, 21 individual cells; and siPERK+PERK-ΔLD-3SA, 18 individual cells). **(B)** Requirement of PERK for dense parallel cristae formation mitochondria in interscapular BAT (iBAT). iBAT was fixed and analysed by electron microscopy. The mitochondria were divided into mitochondria possessing dense parallel cristae and mitochondria possessing sparse cristae and were counted in electron micrographs. The results are shown as described in Fig 2E. Data are shown as the average from 20–30 ($PERK^{+/+}$ mice) or 22–33 ($PERK^{-/-}$ mice) electron micrographs ($n$ = 3 independent experiments). **(C)** Expression of mitochondrial and ER proteins in iBAT from newborn $PERK^{+/+}$ or $PERK^{-/-}$ mice. Lysates from iBAT were analysed by immunoblotting with the indicated antibodies ($n$ = 8 mice). **(D, E, F, G)** Quantification of uncoupling protein 1 (D), COX4 (E), Cyt C (F), and Tom20 (G) band intensity (see Fig 5C). Uncoupling protein 1, COX4, and Cyt C were normalized to Tom20. Tom20 was normalized to actin. Data are shown as the fold change relative to the average band intensity of $PERK^{+/+}$ mice ($n$ = 8 independent individuals). **(H)** Representative thermographic images and dorsal interscapular skin temperature of $PERK^{+/+}$ or $PERK^{-/-}$ newborn mice during exposure to 16°C. The newborn mice, which were incubated at 32°C, were placed on a 16°C plate (0 min). The back skin temperature was measured by an infrared thermographic camera. $P$ = 0.01037 ($PERK^{+/+}$, $n$ = 13; $PERK^{-/-}$, $n$ = 8.) **(A, D, E, F, G, H)** Data information: data are presented as mean ± SEM. NS; *$P$ < 0.05; **$P$ < 0.01; ****$P$ < 0.0001 (repeated measures ANOVA (A, H), $t$ test (D, E, F, G)).

mitochondrial OXPHOS uncoupler carbonyl cyanide m-chlorophenyl hydrazine (CCCP), endogenous PERK was recognized by PERK-3S[P] Ab (Fig S6C). Moreover, PERK was required for CCCP-induced thermogenesis in BAs (Fig S6D). Thus, we hypothesize that the secondary events triggered by mitochondrial stress, e.g., generation of reactive oxygen species, influx/efflux of calcium ions, and loss of ΔΨm may activate the PERK–GABPα axis. However, the fact that the antioxidant reagent N-acetyl cysteine did not inhibit PERK phosphorylation induced by CCCP suggests that reactive oxygen species may not mediate the transduction of mitochondrial stress signals to PERK (data not shown). Perturbation of the cellular lipid composition activated PERK and IRE1α independently of the UPR (Promlek et al, 2011; Volmer et al, 2013), whereas the phosphorylation status of IRE1α was not affected on day 2 (Fig 2A and B). We, thus, speculate that PERK phosphorylation might not be triggered by alterations in lipid composition. Identification of the kinase that is activated during BA differentiation and that contributes to PERK phosphorylation may clarify the detailed mechanism by which mitochondrial inner membrane protein biogenesis is regulated in BAs.

Another important question is how PERK regulates mitochondrial inner membrane protein biogenesis and crista formation. Although PERK deficiency had no effect on the areas of ER–mitochondria

contact sites (Fig S3H–J), PERK enrichment in the MAM might contribute specifically to parallel crista formation in ER-attached mitochondria (Verfaillie et al, 2012). Because we do not yet have evidence regarding the physiological relevance of the increased areas of ER–mitochondria contact sites to PERK–GABPα axis-mediated mitochondrial function, further investigation is necessary. The mRNA expression of Ppargc1a, which encodes the thermogenic coactivator PGC-1α (Kang et al, 2005), was lower in PERK-deficient BAs than in control BAs (Fig 6A). PGC-1α binds GABPβ1 and transcriptionally activates GABPα target genes associated with mitochondrial respiratory complex genes (Wu et al, 1999; Handschin et al, 2007). Because Ppargc1a gene expression is regulated by several transcription factors, including ATF2 (Cao et al, 2004), MEF2 (Handschin et al, 2003), and CREB (Herzig et al, 2001), which are known to be targets of GABPα (chromatin immunoprecipitation sequencing datasets, http://amp.pharm.mssm.edu/Harmonizome/) (Rouillard et al, 2016), we reasoned that Ppargc1a may be one of the primary targets of the PERK–GABPα axis. Because the reductions in Cyt C, COX4, and UCP1 in PERK-deficient BAs were partially ameliorated by exogenously expressed PGC-1α (Fig S6E, lane 6), some mitochondrial genes targeted by PERK may be regulated by the cooperation of PGC-1α with GABPα. Because phosphorylation of GABPα and

GABP$\beta$1 by ERK and JNK contributes to transcriptional activation of GABP$\alpha$ target genes in skeletal muscle cells (Fromm & Burden, 2001), PERK or its downstream target kinase may phosphorylate and regulate the GABP$\alpha$ pathway. Another possibility is that abnormal crista formation may occur before reduction in the expression of mitochondrial inner membrane and crista proteins given that there is a correlation between crista formation and mitochondrial inner membrane protein biogenesis. Crista morphology is maintained by cardiolipin synthesis and the formation of crista junctions through the mitochondrial cristae organizing system (MICOS) (Pfanner et al, 2014; Guarani et al, 2015; Anand et al, 2016; Kojima et al, 2019). In BAs and beige adipocytes, cardiolipin synthesis is essential for systemic energy homeostasis mediated by thermogenesis (Sustarsic et al, 2018). Interestingly, our RNA sequencing analysis of HEK293 cells revealed that *MIC10*, which encodes one of the MICOS components, was reduced by PERK deficiency (Table S2), and analysis of chromatin immunoprecipitation sequencing datasets (Harmonizome) revealed that some MICOS component genes, including *MIC10*, *MIC26*, *MIC60*, and *MIC19*, are targets of GABP$\alpha$ (Rouillard et al, 2016). One possible explanation is that defective crista junction formation initially triggered by inhibition of MIC10 may attenuate the expression of mitochondrial inner membrane and crista proteins in PERK-deficient BAs, resulting in dysfunction of mitochondria in these BAs.

In conclusion, our findings demonstrate a novel mechanism by which BAs acquire fully developed and functional mitochondria to produce ATP and heat. PERK maintains mitochondrial homeostasis in addition to ER homeostasis through independent mechanisms other than eIF2$\alpha$–ATF4 axis signalling. Although further investigation is needed to clarify the mechanisms by which PERK is phosphorylated at Ser715, Ser717, and/or Ser719 during BA differentiation, PERK–GABP$\alpha$ axis signalling may be a target through which to increase energy metabolism without affecting ER homeostasis.

# Materials and Methods

## Cell culture

The primary stromal vascular fraction from iBAT of newborn ICR mice was obtained by collagenase digestion. The digested iBAT was filtered through a 100-$\mu$m nylon cell strainer, and the cells were isolated by centrifugation (156$g$) for 5 min. The cell pellet was washed with PBS and cultured with DMEM supplemented with 20% FBS. The cells were seeded, grown to confluence (designated day 2) and cultured with differentiation induction medium (DMEM [Sigma-Aldrich] supplemented with 20% FBS, 20 nM insulin [Sigma-Aldrich], 1 nM triiodo-L-thyronine [T3; Sigma-Aldrich], 5 $\mu$M dexamethasone, 0.125 mM indomethacin [Sigma-Aldrich], 0.5 mM IBMX [Sigma-Aldrich], and 1 $\mu$M rosiglitazone [Sigma-Aldrich]) on day 0. After differentiation induction, the cells were cultured with differentiation enhancement medium (DMEM supplemented with 20% FBS, 20 nM insulin, and 1 nM T3) on days 2 and 4. HEK293 cells were maintained in DMEM containing 10% FBS and penicillin–streptomycin (glycolysis conditions). For habituation to OXPHOS conditions, HEK293 cells were cultured with glucose-free DMEM (Nacalai tesque) containing 10% FBS, 1 mM sodium pyruvate, 10 mM galactose and penicillin–streptomycin and incubated for 12 or 24 h.

## Mice

ICR mice (SLC) were raised under specific pathogen-free conditions at the animal facility of University of Miyazaki. The *PERK*$^{-/-}$ mice have been previously described (Zhang et al, 2002). Newborn *PERK*$^{-/-}$ mice were obtained after mating *PERK*$^{+/-}$ male and female mice. The *PERK*$^{+/-}$ mice were maintained under specific pathogen-free conditions at the Institute of Genome Research of Tokushima University. All animal experiments were approved by the University of Miyazaki and Tokushima University and were performed in accordance with the appropriate institutional guidelines.

## Plasmid and siRNA transfection

AARE-luc plasmid has been previously described (Miyake et al, 2016). A GABP$\alpha$-binding element luciferase promoter reporter construct (GABP$\alpha$-luc) was provided by Dr Nono Tomita (University of Tokyo, Tokyo, Japan) (Hayashi et al, 2007). pRL-CMV was purchased from Promega. The cells were transfected using FuGENE6 (Roche) or polyethylenimine (PEI)-Max (Polysciences). siRNA transfection was performed using Lipofectamine RNAiMAX reagent (Invitrogen). siRNAs (mouse PERK, MSS203819, MSS203821, and NM_010121.2_stealth_3673; mouse GABP$\alpha$, #1-MSS274443, and #2-MSS274444; mouse IRE1$\alpha$, #1-MSS234443, and #2-MSS234445; and mouse ATF6, #1-MSS213140, and #2-MSS279117) were purchased from Invitrogen. Negative Control Med GC Duplex was used as the control. The PERK siRNA (siPERK) MSS203821 was used in Figs 3A–F, 4A–E, 6A–C, 7A, S3A–J, S5E and F, and S6E. The siPERK MSS2023819 was used in Figs 5B and C, 5E–G, 7C and D, 8A, S4A and B, and S6D. NM_010121.2_stealth_3673 was used in Fig 5H. siRNAs are listed in Table S3.

## Generation of retrovirus and infection

PERK-$\Delta$LD-Flag, PERK-$\Delta$LD-KA-Flag, PERK-$\Delta$LD-3SA-Flag, PERK-Flag, PERK-KA-Flag, PERK-3SA-Flag, Flag-GABP$\alpha$, Flag-GABP$\beta$1, and PGC-1$\alpha$-Flag were constructed into the pMXs-IP plasmid. Recombinant retroviruses were produced by transient transfection of retroviral packaging cells (Platinum-GP) using PEI-Max, and primary brown preadipocytes were infected with recombinant retroviruses in DMEM containing 7.5 $\mu$g/ml polybrene (Sigma-Aldrich) for 24 h.

## Generation of CRISPR KO and knock-in cells

PERK-KO or PERK-Flag-knock-in HEK293 cells were generated using the CRISPR/CRISPR-associated 9 (Cas9) system. gRNAs for KO (5′-GATCCTGTTCTTCTTTTACAC-3′ or 5′-GACTGCAATTATGCTATCAAG-3′) and knock-in (5′-AGCAATTAGCCTTAAGTTGT-3′ or 5′-GTTGTGCTAGCAACCC-TAAT-3′) were ligated into pX330 or pX335 vectors (Addgene). To generate knock-in cells, 5′ arm and 3′ arm sequences were amplified from genomic DNA by PCR using 5′ arm primers (5′-TTCAGCC-TAATGTCCAGTGT-3′ and 5′-ATTGCTTGGCAAAGGGCTA-3′) and 3′ arm primers (5′-CTTCTTAGAATATGCCTGTC-3′ and 5′-ATGACTCTTCCTA-GATCTA-3′). These arm sequences were inserted together with 3× Flag, Poly A, $\beta$ actin promoter, and Neo sequences in the empty

vectors, which were used as donor vectors. The plasmids were transfected into HEK293 cells using PEI-Max, and single cells were seeded in a 96-well plate or selected by G418. These clones were analysed by IB.

### Intracellular ATP assay

An ATP assay was performed using an ATP Bioluminescence Assay Kit CLS II (Roche). The intensity of luminescence was measured with a TD-20/20 luminometer (Promega). The ATP content was normalized to the cell number. The amount of oligomycin A (Sigma-Aldrich)–insensitive intracellular ATP was used to calculate glycolysis-dependent ATP production. OXPHOS-dependent ATP production was calculated from the amounts of total intracellular ATP and glycolysis-produced ATP (Ojaimi et al, 2002; Yang et al, 2014).

### Measurement of cellular basal OCR and ECAR

Primary brown preadipocytes were seeded in an XFp cell culture plate at 20,000 cells/well and then differentiated. On day 0, 2, 4, and 6, the culture medium was replaced with an XF Assay Medium supplemented 10 mM glucose, 1 mM pyruvate, and 2 mM glutamine. After 60 min incubation at 37°C in $CO_2$-free incubator, the OCR and ECAR were measured using an XFp extracellular flux analyser (Agilent Technologies) and XF Cell Mito stress test kit, and then the basal OCR and energy phenotype profiling (basal OCR/ECAR) were calculated. The mitochondrial stress test was performed according to the manufacturer's instructions (8 $\mu$M oligomycin A, 1 $\mu$M carbonyl cyanide-p-trifluoromethoxyphenylhydrazone, 1 $\mu$M rotenone, and 1 $\mu$M antimycin A). The data were normalized by total protein content.

### Luciferase assay

siRNA-transfected brown preadipocytes were co-transfected with 10 $\mu$g of AARE-luc or GABP$\alpha$-luc and 1 $\mu$g of Renilla-luc plasmids and infected with retroviruses expressing Venus, PERK-Flag, PERK-KA-Flag, PERK-3SA-Flag, PERK-ΔLD-Flag, PERK-ΔLD-KA-Flag, or PERK-ΔLD-3SA-Flag. A luciferase assay was performed using a Dual-Luciferase Reporter Assay System (Promega). The promoter activity was normalized using Renilla reporter values for transfection efficiency according to the manufacturer's protocol.

### IB and immunoprecipitation

IB and immunoprecipitation experiments have been described in detail (Kadowaki et al, 2015). Cells and iBAT were lysed in lysis buffer (20 mM Tris–HCl, pH 7.5, 150 mM NaCl, 5 mM EGTA, 1% Triton X-100, $\beta$-glycerophosphate, and 1% sodium deoxycholate) containing 1 mM phenylmethylsulfonyl fluoride, 1 mM $Na_3VO_4$, and 50 mM NaF on ice. The proteins were separated by SDS–PAGE, blotted onto PVDF membranes and blocked in TBS/T containing 5% dry milk. The membranes were incubated with antibodies and detected by an ECL system. The immunoprecipitation experiments were performed using an anti-Flag M2 antibody affinity gel (Sigma-Aldrich). After washing the gels, the immunoprecipitates were detected by IB. Band intensity was measured by ImageQuant TL (GE Healthcare) or ImageJ software (National Institutes of Health). Antibodies are listed in Table S3.

### λ phosphatase treatment

Cell lysates were treated with 2 units of $\lambda$ phosphatase (New England BioLabs) at 30°C for 30 min. SDS sample buffer was added, and the samples were incubated at 98°C for 5 min.

### Puromycin labeling assay

Cells were treated with 10 $\mu$g/ml puromycin for 10 min, immediately washed with PBS and lysed. The puromycin-labeled proteins were analysed by IB with an anti-puromycin antibody. Antibodies are listed in Table S3.

### Quantitative PCR analysis

Total RNA was isolated from cells using RNAiso (Takara). RT was performed using RevaTra Ace qPCR RT Master Mix with gDNA Remover (TOYOBO). qPCR was performed with a StepOnePlus Real-Time PCR System (Applied Biosystems) with SYBR Green PCR Master Mix (Kapa Biosystems). The expression level of *S18* mRNA was used as a normalization control. The primer sequences are shown in Table S3.

### Mitochondrial DNA quantification

Cells were digested with proteinase K for 2 h at 55°C, and total DNA was extracted with phenol/chloroform. Quantification of genomic DNA ($\beta$-globin) and mtDNA (COX2) was performed using qPCR. The primer sequences are provided in Table S3.

### RNA sequencing

Control and PERK-deficient differentiating cells were cultured for 16 or 24 h in differentiating enhancement medium on day 2. WT and PERK-deficient HEK293 cells were incubated with OXPHOS medium for 12 h. Total RNA was isolated using an RNeasy kit (Invitrogen), and the quality (RNA integrity number: 10) was measured using a Bioanalyser 2100 (Agilent Technologies). Purification of mRNA was performed using NEBNext poly (A) magnetic beads (New England BioLabs). DNA libraries were prepared with an NEBNext DNA Library Prep Master Mix Set for Illumina and NEBNext Singleplex Oligos for Illumina (New England BioLabs) according to the manufacturer's instructions. The sequence reads from the RNA sequencing analysis were mapped to the human reference genome (GRCm38/mm10 or GRCh37/hg19) using TopHat version 2. The mapped sequences were converted to expression levels (fragments per kilobase of exon per million reads mapped, FPKM) using Cufflinks and Cuffdiff. The results are expressed as the log2 fold change relative to the levels in control differentiating cells or WT HEK293 cells. Gene Ontology analysis was performed using PANTHER (http://geneontology.org). A binding motif assay was performed by MEME motif analysis (http://meme.sdsc.edu) (Bailey et al, 2009).

## LC-MS/MS–based phosphoproteomic analysis and sample preparation

To identify the phosphorylation sites of PERK in differentiating cells, siPERK-transfected primary brown preadipocytes were infected with retroviruses expressing Flag-tagged PERK-ΔLD-KA, and extraction was performed after 12 h of culture with differentiation enhancement medium on day 2. PERK-ΔLD-KA was purified from the cell lysates using M2-Flag beads and eluted with 3× Flag peptides. The Purified PERK-ΔLD-KA was digested with Glu-C (Promega) at 37°C for 18 h. The peptides were desalted with a Monospin C18 column (GL Sciences), and then the phosphopeptides were enriched using a High-Select Fe-NTA Phosphopeptide Enrichment Kit (Thermo Fisher Scientific). The phosphopeptides were analysed by nanoscale LC-MS/MS using an Ultimate300 RSLCnano (Thermo Fisher Scientific) and a Q-Exactive mass spectrometer (Thermo Fisher Scientific). The LC separation was performed using an EASY-Spray column (75-$\mu$m inner diameter, 25 cm packed with C18 reversed-phase resin). The MS/MS was performed in a data-dependent fashion with a top 10 method. For peptide mapping and identification of phosphorylation sites, raw data were analysed using Proteome Discoverer and searched for peptides containing phosphoserine, phosphothreonine, or phosphotyrosine against the MASCOT and SEQUEST HT search engine.

## Immunofluorescence staining

Endogenous ER proteins and mitochondrial proteins were detected by anti-KDEL and anti-Tom20 antibodies. On sterile glass coverslips, primary brown preadipocytes were cultured and differentiation was induced, and the cells were harvested on day 6. After fixing in 4% paraformaldehyde at room temperature for 25 min, the cells were washed with PBS, permeabilized in 0.2% Triton X-100, blocked with 1% BSA in PBS for 1 h, and then incubated with primary antibodies for 12 h at 4°C. After washing with PBS, the cells were incubated with Alexa Fluor 488– or Alexa Fluor 546—conjugated secondary antibodies (Thermo Fisher Scientific) for 1.5 h at room temperature. The cells were washed with PBS and mounted in VECTASHIELD mounting medium containing DAPI (Vector Laboratories). Immunofluorescence images were obtained using fluorescence microscopy and confocal laser microscopy (TSC-SP8; Leica). Quantitative analysis of the cell images was performed using ImageJ software. Antibodies are listed in Table S3.

## Electron microscopy

Cells and iBAT were washed with PBS and fixed in modified Karnovsky's fixative (3% glutaraldehyde and 1.6% paraformaldehyde in 0.1% sodium cacodylate, pH 7.4) before being fixed in 1% osmium tetroxide. After dehydration with an ethanol series (50%, 70%, 80%, 90%, 95%, and 100%), ultrathin sections (70–90 nm) were cut, stained, with uranyl acetate and imaged using an HT7700 transmission electron microscope (HITACHI) at 100 kV. ER–mitochondria contact sites were defined as sites with a distance of <30 nm between the membranes of the two organelles. Quantification of the ER–mitochondria distance and contact length was performed using ImageJ software.

## Oil red O staining

Primary BAs were fixed with 10% formalin in PBS for 10 min, washed with PBS, rinsed with 60% isopropanol, and stained with oil red O solution. After washing with 60% isopropanol and PBS, images were obtained. To quantify the amounts of LDs, oil red O (Sigma-Aldrich)–stained cells were lysed in 100% isopropanol containing Nonidet P-40 for 10 min. Absorbance of isolated oil red O was determined by a spectrophotometer (492 nm wavelength).

## ΔΨm assay

For the ΔΨm assay, the cells were incubated with MITO-ID Membrane Potential Detection Reagent and Necrosis Detection Reagent (Enzo Life Sciences) for 15 min at room temperature and analysed by confocal fluorescence microscopy or flow cytometry (BD FACSVerse; BD Biosciences).

## Intracellular thermogenesis assay

An intracellular thermogenesis assay was performed using two different thermoprobes. BAs (day 6) were washed with PBS and 5% glucose solution and then incubated with 0.04 wt/vol% cellular thermoprobe (FDV-0004; Funakoshi) for 10 min. The fluorescence intensity of single cells was observed before and after stimulation with 0.5 $\mu$M CL316,243 (Sigma-Aldrich) by confocal laser microscopy (TSC-SP8; Leica). A hydrophilic fluorescent nanogel thermometer (Gota et al, 2009) was microinjected into the cytosol of BAs. The fluorescence intensity of single cells was observed before and after stimulation with 10 $\mu$M CCCP (Sigma-Aldrich) under an IX70 inverted microscope (Olympus) equipped with an objective lens (60×, UplanApo N.A. 1.40; Olympus). A cooled charge-coupled device camera (ORCA-ER) was used to acquire cell images.

## Thermographic analysis

Newborn mice were incubated at 32°C and placed on a 16°C plate, and thermographic images were obtained with an infrared thermographic camera (TVS-200; Nippon Avionics). Thermographic analysis was performed using PE Professional (GORATEC).

## Histopathological analysis

Tissue sections of 4-$\mu$m thickness were stained with haematoxylin and eosin, and images were obtained by microscopy (DM1000 LED; Leica).

## Quantification and statistical analysis

The data are presented as the mean ± standard error, and statistical analysis was performed using $t$ test and repeated measures ANOVA to compare data in different groups. Statistical analysis was performed by using EZR software version1.30 (Kanda, 2013).

## Data and software availability

RNA-seq data used in this study have been deposited in the NCBI Gene Expression Omnibus under accession number GSE113572 and GSE132598. IB experiments were repeated at least three times independently, and the representative data are shown. All datasets are available from the corresponding author upon reasonable request.

## Supplementary Information

## Acknowledgements

We thank N Tomita (University of Tokyo) and K Mikoshiba (RIKEN) for providing us with plasmids. We also thank K Imaizumi (Hiroshima University), M Nakamura Tokyo Medical and Dental University (TMDU), N Ishihara (Osaka University), A Sawaguchi, Y Goto, A Nakatake, and laboratory members (University of Miyazaki) for technical assistance and discussion. This study was supported by AMED (Japan Agency for Medical Research and Development) (grant number JP19gm5010001) (H Ichijo), Ministry of Education, Culture, Sports, Science and Technology (MEXT) KAKENHI (grant number JP18H03995 [H Ichijo], 18K06916 [H Kato], 17H06419 [H Nishitoh], 18H02973 [H Nishitoh], and 18H04699 [H Nishitoh]) from the Japan Society for the Promotion of Science, Mitsubishi Foundation, Uehara Memorial Foundation, Astellas Foundation for Research on Metabolic Disorders, Takeda Science Foundation, Project for Creation of Research Platforms and Sharing of Advanced Research Infrastructure, and Nanken-Kyoten, TMDU (to H Nishitoh).

## Author Contributions

H Kato: conceptualization, funding acquisition, investigation, and writing—original draft.
K Okabe: resources and investigation.
M Miyake: resources.
K Hattori: methodology.
T Fukaya: resources and investigation.
K Tanimoto: resources and investigation.
S Beini: investigation.
M Mizuguchi: investigation and methodology.
S Torii: methodology.
S Arakawa: methodology.
M Ono: investigation and methodology.
Y Saito: methodology.
T Sugiyama: software and formal analysis.
T Funatsu: supervision and methodology.
K Sato: supervision and methodology.
S Shimizu: supervision and methodology.
S Oyadomari: resources.
H Ichijo: supervision and methodology.
H Kadowaki: conceptualization, supervision, and writing—review and editing.
H Nishitoh: conceptualization, data curation, supervision, funding acquisition, project administration, and writing—original draft, review, and editing.

## Conflict of Interest Statement

The authors declare that they have no conflict of interest.

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
