## [Reviewer comments · Life Science Alliance]

Life Science Alliance

ER-resident sensor PERK is essential for mitochondrial thermogenesis in brown adipose tissue

Hironori Kato, Kohki Okabe, Masato Miyake, Kazuki Hattori, Tomohiro Fukaya, Kousuke Tanimoto, Shi Beini, Mariko Mizuguchi, Satoru Torii, Satoko Arakawa, Masaya Ono, Yusuke Saito, Takashi Sugiyama, Takashi Funatsu, Katsuaki Sato, Shigeomi Shimizu, Seiichi Oyadomari, Hidenori Ichijo, Hisae Kadowaki, and Hideki Nishitoh

DOI: <https://doi.org/10.26508/lsa.201900576>

Corresponding author(s): *Hideki Nishitoh, University of Miyazaki*

Review Timeline:

Submission Date:	2019-10-09
Editorial Decision:	2019-10-09
Revision Received:	2020-01-09
Editorial Decision:	2020-01-21
Revision Received:	2020-01-22
Accepted:	2020-01-23

Scientific Editor: Andrea Leibfried

Transaction Report:

Please note that the manuscript was previously reviewed at another journal and the reports were taken into account in the decision-making process at Life Science Alliance.

Referee #1 Review

Report for Author:

In this manuscript Kato and co-authors investigate the role of the ER kinase, PERK, in modulating mitochondrial biogenesis. In line with several previous studies, the authors show that increased ER-mitochondria contact is associated with mitochondrial biogenesis. The authors provide novel insight into the role of ER in regulating mitochondrial function and development during brown adipocyte differentiation. Considering the unique cellular organization of brown adipocytes with high mitochondrial and lipid content and extremely high metabolic flux, yet relatively scarce levels of ER content/surface, the brown adipocyte differentiation is an interesting model system. The authors

show that PERK phosphorylation by unknown kinases play an important role in brown adipocyte mitochondrial biogenesis and that the downstream signals are independent of UPR and at least in part mediated by GABPa. This is an interesting finding; however, the manuscript unfortunately lacks mechanistic insight into how PERK is activated and how phosphorylation of PERK signals to GABPa. Overall the manuscript is very speculative, and a major part of the discussion is devoted to speculation about these up- and downstream signals.

Major points

- 1) There are several claims and speculations in the manuscript that are not sufficiently supported. E.g. at the end of the introduction the authors state that "the molecular mechanism of mitochondrial stress-induced PERK activation may be a target through which to increase energy metabolism without affecting ER homeostasis." The authors write "Identification of the kinase that is activated by mitochondrial stress and contributes to PERK phosphorylation may clarify the detailed mechanism by which mitochondrial biogenesis is regulated in BAs." What is the evidence that it is mitochondrial stress activates the PERK-GABPa axis? It might as well be a non-mitochondrial signal. Similarly, the involvement of PGC1a in mediating the response is also purely speculative.
- 2) The claim that PERK is physiologically phosphorylated independently of ER stress / UPR should be more carefully investigated. Basic signaling markers of the pathway should be investigated such as the ER stress sensor activity, the transcription factors regulated and some key UPR gene targets. The authors assayed PERK phosphorylation, ATF6 activity (indirectly through GRP78), and ATF4 (Chop and Gadd34) induction. However, for IRE1a signaling the analysis is incomplete. Firstly, the authors use a total IRE1a antibody to comment on the phosphorylation status of IRE1a; secondly, the authors do not investigate downstream targets of the IRE1a pathway (e.g. Xbp1 splicing). A careful investigation of all UPR signaling pathways is necessary to confirm that PERK is specifically activated independently of other UPR pathways, which is a major claim of the manuscript.
- 3) The authors state that PERK was temporarily phosphorylated after 6h to 12h of culture with differentiation enhancement medium on day 2. How can the authors be sure that this phosphorylation is not induced by the medium change itself which causes cellular stress? The authors should perform a detailed time course throughout the BA differentiation to make sure the temporal phosphorylation is not induced by media change per se.
- 4) The link between PERK activation and GABPa remains elusive. The authors must strengthen the mechanistic insight into how PERK activates GABPa. Furthermore, the bioinformatic analyses leading to the GABPa prediction is poorly documented.
- 5) The observation that only some mitochondria possessed abnormal cristae in PERK-deficient brown adipocytes is interesting. Do the mitochondria with abnormal cristae have more contact sites to ER than mitochondria with a "normal" morphology in PERK-deficient BAs?
- 6) Too many important figures are placed in supplementary throughout the paper which makes it difficult to read the manuscript.

Minor comments:

- 1) Quantification of Figure EV2A would greatly strengthen the conclusion that the amounts of

PERK decreased during BA differentiation.

2) To complement Figure 1A, a figure illustrating the total area/volume of the ER and the mitochondria throughout the differentiation is a good addition.

3) The term "DNA-mediated transcription" is inappropriate, it should just be "transcription".

4) The statement in the introduction "Thermogenesis in brown adipocytes (BAs) is mediated by the function of uncoupling protein 1 (UCP1), which localizes to the mitochondrial inner membrane and dissipates the mitochondrial proton electrochemical gradient via β 3-adrenoceptor (β 3AR) stimulation." Is not correct. The β 3 receptor induces but does not mediate the dissipation of the proton electrochemical gradient.

Referee #2 Review

Report for Author:

The manuscript of Kato and collaborators, explores the role of PERK in mitochondria biogenesis and bioenergetics during brown adipocytes (BA) differentiation. The authors show that during BA differentiation PERK is phosphorylated independently of its UPR function, by a yet to identified protein kinase(s). This leads to transcriptional activation of GABPa, which regulates mitochondrial biogenesis. Finally, the authors show that and physiological PERK phosphorylation and the PERK-GABPa axis is involved in thermogenesis in BAT. The results are interesting and unravel a potential and novel mechanism of PERK regulation of BAT, but there are several issues authors need to address, including apparently ignored signaling mechanisms, in order to be more convincing.

Major points:

Fig. 1A: Panel B: Is the increased ATP production through OXPHOS significant? Also, it would be mandatory to couple this indirect measurement of mitochondrial ATP production, to actual measurements of oxygen consumption rate.

Fig 1E: The surface of the ER-mitochondria contact sites is increased, likely as a result of the increased mitochondria surface induced during differentiation. However, given that the ER surface (and likely the volume too) is also reduced, one would expect that the volume of the contact sites is reduced as well. Can the authors comment on this?

There should be a more detailed quantification in terms of the organellar changes, which seem drastic (why is the ER so dramatically reduced?), usually ER-mitochondria contact length should be normalized to mitochondrial perimeters (or number). It would be also important to see a bigger field of view of the EM, not just a zoom. How does the overall cell look like? Beyond morphology, do the mitochondria change in number and subcellular distribution?

-The authors state that the cells are adapting to enable a metabolic shift, but the amount of ATP coming from mitochondria ("OXPHOS") on day 6 vs day 2 in panel B is basically identical but if you look at panel D, there is a vast shift in the amount of mitochondrial proteins. On day 2 there are still only a tiny fraction of mitochondrial proteins visible (also overall much less ATP anyway). The authors claim that the increase in mitochondria is due to the cell's shift from glycolysis to more OXPHOS but these data do not completely support that, rather the major change seem to be accounted by a decrease in the glycolytic contribution to ATP. Is this accompanied by a change in glucose uptake and utilization?

Fig. 2A implies that PERK phosphorylation causes the shift in the molecular weight of the protein. Since these are murine cells, the authors should use anti-P-PERK antibody (commercially available and validated) to analyze whether the autophosphorylation site, and therefore the active form of PERK, is evoked during differentiation. Since the use of PERK kinase dead mutant highlight that the PERK kinase activity is required for many (if not all) the effects observed, this analysis is required.

Fig. 2D: In the EM analysis the authors show in this panel, clear differences in the size and elongated morphology of the mitochondria, beyond crista differences, are visible in the PERK silenced cells (quite visible also in EV2). Moreover, in 2F silencing PERK seems to impact all mitochondria proteins including tom20, can the authors comment on this and perhaps expand the analysis to other outer mitochondria proteins. Without a clear explanation of these effects the following conclusion; "These observations are consistent with the result that PERK deficiency affects the number of cristae but not the total area of mitochondria (Figs 2E and EV2I)" is muddled.

-Are there differences in the levels of fusogenic proteins, like OPA1, which could explain also the crista phenotype in PERK deficient cells? Does absence of PERK affect mitochondria fusion and fission events? or mitophagy? These points need to be at least discussed.

-Can the authors also explain why in Fig 2F, the ER associated proteins -especially GRP94- show no changes till day 6, while in Fig. 1D they are reduced already at day 2, leading the authors to conclude that there is a reduced ER area.

EV2 K-L: The representative pictures do not seem to match the quantification; in the siRNA PERK cells the extension of the contact sites (if not the number as well) appear quite reduced. Hence the authors are missing or ignoring some quite interesting data. Contact sites/morphology of ER/Mito is markedly different, between Ctr and siRNA PERK. This analysis deserves to be better characterized, including the analysis of the mitochondria area.

Fig. 3: All of the experiments in Figure 3 are comparing PERK siRNA treated cells re-expressing PERK-DLD to ctr siRNA cells. But there is no comparison with the re-expression of full-length PERK, which is the proper control. Also, unless Venus is co-expressed also the PERK-DLD re-expressing cells, this is not a proper control.

Fig.3A: The authors are using shift in the molecular weight of PERK by WB, as a readout of PERK phosphorylations after adding the differentiation medium but it is not always clear what kind of shift /phosphorylation status the authors mean. It would be helpful to have arrows pointing this out. All the different bands for PERK-DLD are also very confusing. What do they all mean? Why is there suddenly this big fat band around 100 kDa supposedly showing the phosphorylation signal? Lambda phosphatase treatment should be done on the PERK-DLD samples to deconvolute all the different bands that are shown.

-The mutant PERK- Δ LD-KA does present all the phosphorylation sites for the S115/117/119 (shown in Fig EV3 C and A) but it does not rescue the PERK-mediated mitochondrial biogenesis phenotype, suggesting that the kinase activity is fundamental, probably together with the S115, 117 and 119. But the authors seem to ignore this and conclude "Collectively, our observations suggest that Ser715, Ser717 and/or Ser719 of PERK are indispensable for mitochondrial biogenesis and functions but not for the UPR".

Fig. 4A: Student's t-test is the wrong statistical test for this kind of data with multiple comparisons /samples in one graph. The authors should use appropriated statistical tests like ANOVA in this

case and through the manuscript. The authors use student's t-test nearly ubiquitously but incongruently. Examples of misuse are in figures 4A and EV5D, but also why are data in Figure 5A analyzed using ANOVA, but an identical dataset in Fig EV5D with a t-test?

Fig. 4D: Also in this panel the authors show that both the PERK 3SA and the PERK KA mutants do not rescue GABP α transcriptional activity, indicating that the kinase activity of PERK is important. Hence the conclusion the author draw based on these results: " suggesting that PERK specifically regulates the GABP α pathway as a result of PERK phosphorylation at Ser715, Ser717 and/or Ser719 during BA differentiation." should be revisited to include the possibility that also the kinase activity

-A major conclusion from the data of Fig. 4 is that PERK phosphorylation/kinase activity is mechanistically linked to GABP α activation, which in turn regulates the expression of mitochondrial genes and mito respiration, but it would be important to show that GABP α regulates mito cristae morphology. It remains to be clarified how does PERK cause the activation of GABP α . Why does it need its kinase activity -if it is not UPR dependent- and why does it need to be phosphorylated?

Fig. 5A: The authors should also include data showing the effects of the kinase dead mutant (PERK- Δ LD-KA) on the thermogenic defect. Actually, since the PERK- Δ LD-3SA mutant only partially rescues it, again here there exists the possibility that PERK kinase activity is involved. Moreover, why doesn't the mutant PERK- Δ LD rescue completely the phenotype?

Minor points:

2C: Authors need more data to claim the lipid droplet content is unchanged. At least some microscopic staining should be shown.

EV2C: in the same blot please also show the increased P-PERK (taking along also the condition of 6H and 12h of medium incubation where in the figure 2A you show P-PERK) and a positive control of TG/TM at different time points especially because ATF4 signal is not that clear here. If possible eIF2a-P /eIF2a levels should be shown as well, as eIF2a is a direct target of PERK.

EV2G-I: The immunostaining of the mitochondria and the quantification need improvements.

Referee #3 Review

Report for Author:

Kato et al. propose that PERK is essential for BAT thermogenesis. However, this hypothesis is not adequately supported by the data provided.

1. Regulation of PERK phosphorylation during brown adipocyte differentiation is weak. The authors only observe a modest increase in PERK phosphorylation during a short period of the differentiation process (after 6 and 12 hours on day 2). Furthermore, it is unclear whether this increase is actually attributable to the differentiation process or is simply due to the fact that the medium was changed, thereby replenishing various nutrients and growth factors. In the absence of robust regulation of PERK phosphorylation during brown adipocyte differentiation, it is unclear whether PERK phosphorylation is truly relevant in this context or if manipulating PERK levels simply has non-specific effects on mitochondria.

2. The proposed mechanism is also inconsistent with the data provided. The authors show that ER-mitochondria contacts are increased in differentiated brown adipocytes and hypothesize that these contacts are important. However, the authors then attribute the effects of PERK to a transcriptional pathway involving GABP α , which is not clearly linked to the ER-mitochondria contact observation. Similarly, the authors propose that PERK regulates mitochondrial biogenesis through GABP α , but then show that PERK knockdown has no effect on mitochondrial area or DNA content. The major defect appears to be in the morphology of the cristae; however, the mechanism responsible for this effect is not explored experimentally, only conceptually in the discussion.

3. Lastly, the in vivo data are unconvincing. PERK is a ubiquitously expressed and generally important protein. A global, developmental PERK knockout mouse is thus not a reliable model for studying intrinsic PERK function in BAT. The body surface temperature difference is also very modest.

October 9, 2019

Re: Life Science Alliance manuscript #LSA-2019-00576-T

Prof. Hideki Nishitoh
University of Miyazaki
5200, Kihara, Kiyotake
Miyazaki 8891601
Japan

Dear Dr. Nishitoh,

Thank you for transferring your manuscript entitled "ER-resident sensor PERK is essential for mitochondrial thermogenesis in brown adipose tissue" to Life Science Alliance. The manuscript was assessed by expert reviewers at another journal before, and the editors transferred the reports to us with your permission.

The reviewers appreciated your data, but would have expected more mechanistic insight into how PERK is activated and how phosphorylation of PERK signals to GABPalpha. Addressing this criticism is not needed for publication in Life Science Alliance, and we would thus like to invite you to submit a revised version of your manuscript, addressing the technical issues raised and those that pertain to lack of support for the existing data. More specifically, please provide a point-by-point response and edit the text / modify data representation according to the referees' requests. In addition, quantification and proper statistical analysis of all experiments are required. Please also address the following specific reviewer points:

- reviewers' concerns pertaining to inconsistencies between the different datasets
- address Point #2 and #3 from referee #1
- determine whether increased ATP production through OXPHOS is significant and measure oxygen consumption rate (referee #2)
- use commercially available anti-PERK antibody to test that autophosphorylation is evoked during differentiation (referee #2)
- address point #1 from referee #3

The typical timeframe for revisions is three months.

Thank you for this interesting contribution to Life Science Alliance. We are looking forward to

receiving your revised manuscript.

Sincerely,

B. MANUSCRIPT ORGANIZATION AND FORMATTING:

Followings are the point-by-point responses to the reviewers' comments on our previous manuscript submitted to the other journal. However, in our revised manuscript to the *Life Science Alliance*, not all of these comments have been addressed with experimental evidence.

Reviewer #1:

In this manuscript Kato and co-authors investigate the role of the ER kinase, PERK, in modulating mitochondrial biogenesis. In line with several previous studies, the authors show that increased ER-mitochondria contact is associated with mitochondrial biogenesis. The authors provide novel insight into the role of ER in regulating mitochondrial function and development during brown adipocyte differentiation. Considering the unique cellular organization of brown adipocytes with high mitochondrial and lipid content and extremely high metabolic flux, yet relatively scarce levels of ER content/surface, the brown adipocyte differentiation is an interesting model system. The authors show that PERK phosphorylation by unknown kinases play an important role in brown adipocyte mitochondrial biogenesis and that the downstream signals are independent of UPR and at least in part mediated by GABPa. This is an interesting finding; however, the manuscript unfortunately lacks mechanistic insight into how PERK is activated and how phosphorylation of PERK signals to GABPa. Overall the manuscript is very speculative, and a major part of the discussion is devoted to speculation about these up- and downstream signals.

Re: We thank the reviewer for the thoughtful comment and agree with the reviewer's concern that our manuscript lacks mechanistic insight for the upstream and downstream effects of PERK on mitochondrial function in BAs. We are currently trying to address these points, but we have unfortunately not succeeded yet. Our revised manuscript has been toned down with regard to several points, as described in the cover letter to the editor.

Major points

1) *There are several claims and speculations in the manuscript that are not sufficiently supported. E.g. at the end of the introduction the authors state that "the molecular mechanism of mitochondrial stress-induced PERK activation may be a target through which to increase energy metabolism without affecting ER homeostasis." The authors write "Identification of the kinase that is activated by mitochondrial stress and contributes to PERK phosphorylation may clarify the detailed mechanism by which mitochondrial biogenesis is regulated in BAs." What is the evidence that it is mitochondrial stress activates the PERK-GABPa axis? It might as well be a non-mitochondrial signal. Similarly, the involvement of PGC1a in mediating the response is also purely speculative.*

Re: Our response to this comment is described in the cover letter to the editor.

2) *The claim that PERK is physiologically phosphorylated independently of ER stress / UPR should be more carefully investigated. Basic signaling markers of the pathway should be investigated such as the ER stress sensor activity, the transcription factors regulated and some key UPR gene targets. The authors assayed PERK phosphorylation, ATF6 activity (indirectly through GRP78), and ATF4 (Chop and Gadd34) induction. However, for IRE1a signaling the analysis is incomplete. Firstly, the authors use a total IRE1a antibody to comment on the phosphorylation status of IRE1a; secondly, the authors do not investigate downstream targets of the IRE1a pathway (e.g. Xbp1 splicing). A careful investigation of all UPR signaling pathways is necessary to confirm that PERK is specifically activated independently of other UPR pathways, which is a major claim of the manuscript.*

Re: Our response to this comment is described in the cover letter to the editor.

3) *The authors state that PERK was temporarily phosphorylated after 6h to 12h of*

culture with differentiation enhancement medium on day 2. How can the authors be sure that this phosphorylation is not induced by the medium change itself which causes cellular stress? The authors should perform a detailed time course throughout the BA differentiation to make sure the temporal phosphorylation is not induced by media change per se.

Re: Our response to this comment is described in the cover letter to the editor.

4) The link between PERK activation and GABPa remains elusive. The authors must strengthen the mechanistic insight into how PERK activates GABPa. Furthermore, the bioinformatic analyses leading to the GABPa prediction is poorly documented.

Re: We understand the reviewer's concern. We are currently trying to address this point, but we have unfortunately not succeeded yet. Investigation to identify the mechanism by which PERK mediates transcriptional activation by GABP α should be continued.

5) The observation that only some mitochondria possessed abnormal cristae in PERK-deficient brown adipocytes is interesting. Do the mitochondria with abnormal cristae have more contact sites to ER than mitochondria with a "normal" morphology in PERK-deficient BAs?

Re: Our response to this comment is described in the cover letter to the editor.

6) Too many important figures are placed in supplementary throughout the paper which makes it difficult to read the manuscript.

Re: We thank reviewer for the suggestion. Following figures have been placed in main figures.

Previous Fig > Revised Fig:

Fig EV1A > Fig 1A, Fig EV2B > Fig 2D, Fig EV2C > Fig 2E, Fig EV2D > Fig 2F, Fig EV2E > Fig 2G, Fig EV2F > Fig 3A, Fig EV2N > Fig 4A, Fig EV3A > Fig 5A, Fig EV3C > Fig 5C, Fig EV3E > Fig 5D, Fig EV3G > Fig 5H, Fig EV4F > Fig 6D, Fig EV4G > Fig 6E

Minor comments:

1) Quantification of Figure EV2A would greatly strengthen the conclusion that the amounts of PERK decreased during BA differentiation.

Re: We thank the reviewer for the suggestion and quantified the amounts of PERK and IRE1 α during BA differentiation (revised Fig S2).

2) To complement Figure 1A, a figure illustrating the total area/volume of the ER and the mitochondria throughout the differentiation is a good addition.

Re: We thank the reviewer for the suggestion. We showed the data of electron micrographs of BAs during differentiation (revised Fig 1B) and quantified the ER and mitochondrial perimeters (revised Fig 1C and D).

3) *The term "DNA-mediated transcription" is inappropriate, it should just be "transcription".*

Re: We thank the reviewer for the suggestion and corrected as follows.

P4 L4 and P6 L17-18

“nuclear DNA-mediated transcription” > “nuclear DNA transcription”

“mitochondrial DNA-mediated transcription” > “mitochondrial DNA transcription”

4) *The statement in the introduction "Thermogenesis in brown adipocytes (BAs) is mediated by the function of uncoupling protein 1 (UCP1), which localizes to the*

mitochondrial inner membrane and dissipates the mitochondrial proton electrochemical gradient via β 3-adrenoceptor (β 3AR) stimulation." Is not correct. The β 3 receptor induces but does not mediate the dissipation of the proton electrochemical gradient.

Re: We agree with the reviewer's comment and corrected our misleading expressions as follows. P5 L4-7

"Thermogenesis in brown adipocytes (BAs) is mediated by the function of uncoupling protein 1 (UCP1), which localizes to the mitochondrial inner membrane and dissipates the mitochondrial proton electrochemical gradient via β 3-adrenoceptor (β 3AR) stimulation." > "Thermogenesis in brown adipocytes (BAs) is mediated by the function of uncoupling protein 1 (UCP1), which localizes to the mitochondrial inner membrane and dissipates the mitochondrial proton electrochemical gradient."

Reviewer #2:

The manuscript of Kato and collaborators, explores the role of PERK in mitochondria biogenesis and bioenergetics during brown adipocytes (BA) differentiation. The authors show that during BA differentiation PERK is phosphorylated independently of its UPR function, by a yet to identified protein kinase(s). This leads to transcriptional activation of GABPa, which regulates mitochondrial biogenesis. Finally, the authors show that and physiological PERK phosphorylation and the PERK-GABPa axis is involved in thermogenesis in BAT. The results are interesting and unravel a potential and novel mechanism of PERK regulation of BAT, but there are several issues authors need to address, including apparently ignored signaling mechanisms, in order to be more convincing.

Re: We thank the reviewer for the thoughtful comment and agree with the reviewer's concern that our manuscript lacks mechanistic insight for the upstream and downstream effects of PERK on mitochondrial function in BAs. We are currently trying to address this point, but we have unfortunately not succeeded yet. Our revised manuscript has been toned down with regard to several points, as described above.

Major points:

Fig. 1A: Panel B: Is the increased ATP production through OXPHOS significant? Also, it would be mandatory to couple this indirect measurement of mitochondrial ATP production, to actual measurements of oxygen consumption rate.

Re: Our response to this comment is described in the cover letter to the editor.

Fig 1E: The surface of the ER-mitochondria contact sites is increased, likely as a result of the increased mitochondria surface induced during differentiation. However, given that the ER surface (and likely the volume too) is also reduced, one would expect that

the volume of the contact sites is reduced as well. Can the authors comment on this?

Re: We thank the reviewer for the comment and have performed careful quantification and statistical analyses. Organellar changes, including the perimeters of mitochondria and the ER and the lengths of ER–mitochondria contact sites, were quantified from electron micrographs (revised Fig 1C, D, L-N and S3F-J). Although the ER perimeters were decreased and the mitochondrial perimeters increased (revised Fig 1C and D), the ratios of ER–mitochondria contact sites to both the ER perimeters and mitochondrial perimeters were significantly increased (revised Fig 1M and N), suggesting that the ER and mitochondrial membranes may actively contact each other rather than meeting simply because of mitochondrial expansion. These points are commented on the revised manuscript (P8 L14-21).

There should be a more detailed quantification in terms of the organellar changes, which seem drastic (why is the ER so dramatically reduced?), usually ER-mitochondria contact length should be normalized to mitochondrial perimeters (or number). It would be also important to see a bigger field of view of the EM, not just a zoom. How does the overall cell look like? Beyond morphology, do the mitochondria change in number and subcellular distribution?

Re: Our response to this comment is described in the cover letter to the editor.

-The authors state that the cells are adapting to enable a metabolic shift, but the amount of ATP coming from mitochondria ("OXPHOS") on day 6 vs day 2 in panel B is basically identical but if you look at panel D, there is a vast shift in the amount of mitochondrial proteins. On day 2 there are still only a tiny fraction of mitochondrial proteins visible (also overall much less ATP anyway). The authors claim that the increase in mitochondria is due to the cell's shift from glycolysis to more OXPHOS but these data do not completely support that, rather the major change seem to be accounted by a

decrease in the glycolytic contribution to ATP. Is this accompanied by a change in glucose uptake and utilization?

Re: We fully agree with the reviewer's concern. At first, the previous data on ATP production showed the ratio of glycolysis-dependent ATP production and OXPHOS-dependent ATP production in fold changes relative to the total amount of intracellular ATP on day 0 (previous Fig 1B, 2I, 3D, 4E, EV4A and EV4H). Since the total amount of intracellular ATP is different in each stage of brown preadipocytes and BAs because of the different expression levels of UCP1 (revised Fig 1E and K), assessment of the ratio of OXPHOS-dependent ATP production to total ATP production in each cell stage is suitable for assessment of the change in OXPHOS-dependent ATP production capacity during BA differentiation (revised Fig 1F, 4E, 5G, 6G and S5A and F). As shown in the revised Fig 1F, OXPHOS-dependent ATP production in BAs (days 4 and 6) was significantly greater than that in brown preadipocytes (day 2). These data are consistent with the expression of OXPHOS complexes (revised Fig 1J and K). Furthermore, we measured the basal OCR and ECAR, which represent the OXPHOS and glycolytic pathway activity, respectively (revised Fig 1G and H), and found that both were significantly increased during BA differentiation. However, since the increase in the rate of basal OCR was higher than that of ECAR, the basal OCR/ECAR ratio was significantly greater in BAs than in brown preadipocytes (revised Fig 1I). These points are described in the revised manuscript (P7 L20-P8 L7).

Fig. 2A implies that PERK phosphorylation causes the shift in the molecular weight of the protein. Since these are murine cells, the authors should use anti-P-PERK antibody (commercially available and validated) to analyze whether the autophosphorylation site, and therefore the active form of PERK, is evoked during differentiation. Since the use of PERK kinase dead mutant highlight that the PERK kinase activity is required for many (if not all) the effects observed, this analysis is required.

Re: Our response to this comment is described in the cover letter to the editor.

Fig. 2D: In the EM analysis the authors show in this panel, clear differences in the size and elongated morphology of the mitochondria, beyond crista differences, are visible in the PERK silenced cells (quite visible also in EV2). Moreover, in 2F silencing PERK seems to impact all mitochondria proteins including tom20, can the authors comment on this and perhaps expand the analysis to other outer mitochondria proteins. Without a clear explanation of these effects the following conclusion; "These observations are consistent with the result that PERK deficiency affects the number of cristae but not the total area of mitochondria (Figs 2E and EV2I)" is muddled.

Re: Our response to this comment is described in the cover letter to the editor.

-Are there differences in the levels of fusogenic proteins, like OPA1, which could explain also the crista phenotype in PERK deficient cells? Does absence of PERK affect mitochondria fusion and fission events? or mitophagy? These points need to be at least discussed.

Re: We thank the reviewer for the suggestion and agree with the possibility that PERK may regulate mitochondrial fusion/fission and mitophagy, since the importance of mitochondrial dynamics in BAs has been reported by many groups. Although we did not identify a gene related to mitochondrial dynamics from the RNA sequencing data of BAs (Table S1) and HEK293 cells (Table S2), further investigation into whether PERK regulates mitochondrial dynamics is necessary.

-Can the authors also explain why in Fig 2F, the ER associated proteins -especially GRP94- show no changes till day 6, while in Fig. 1D they are reduced already at day 2, leading the authors to conclude that there is a reduced ER area.

Re: Our response to this comment is described in the cover letter to the editor.

EV2 K-L: The representative pictures do not seem to match the quantification; in the siRNA PERK cells the extension of the contact sites (if not the number as well) appear quite reduced. Hence the authors are missing or ignoring some quite interesting data. Contact sites/morphology of ER/Mito is markedly different, between Ctr and siRNA PERK. This analysis deserves to be better characterized, including the analysis of the mitochondria area.

Re: We thank the reviewer for the suggestion. Electron microscopic data in previous Fig EV2K have been revised to Fig S3E, which shows the quantitative data representatively (revised Fig S3H). Moreover, we carefully measured organellar changes, including the perimeters of mitochondria and the ER and the lengths of ER–mitochondria contact sites, from electron micrographs (revised Fig S3F-H). The areas of ER–mitochondria contact sites and the ratios of the contact sites to the ER and mitochondrial perimeters in siPERK BAs were comparable to those in sictrl BAs (revised Fig S3H-J), suggesting that PERK deficiency may not affect ER–mitochondria contact (described on P10 L7-12).

Fig. 3: All of the experiments in Figure 3 are comparing PERK siRNA treated cells re-expressing PERK-DLD to ctr siRNA cells. But there is no comparison with the re-expression of full-length PERK, which is the proper control. Also, unless Venus is co-expressed also the PERK-DLD re-expressing cells, this is not a proper control.

Re: We thank the reviewer for the suggestion. Although we have no retrovirus-encoded PERK full-length now, if necessary, we will generate it and perform the rescue experiment.

Fig.3A: The authors are using shift in the molecular weight of PERK by WB, as a

readout of PERK phosphorylations after adding the differentiation medium but it is not always clear what kind of shift /phosphorylation status the authors mean. It would be helpful to have arrows pointing this out. All the different bands for PERK-DLD are also very confusing. What do they all mean? Why is there suddenly this big fat band around 100 kDa supposedly showing the phosphorylation signal? Lambda phosphatase treatment should be done on the PERK-DLD samples to deconvolute all the different bands that are shown.

Re: We thank the reviewer for the suggestion. The PERK kinase insert region possesses a number of Ser and Thr residues that are highly phosphorylated and required for its activation (Ma et al. Rapid Commun Mass Spectrom 2001, Marciniak et al. JCB 2006). The smear bands at approximately 100 kDa of PERK-ΔLD are thought to be the result of high phosphorylation at the kinase insert region. We are now focusing on BA differentiation-induced phosphorylation and have indicated these bands with white parentheses (revised Fig 5B).

-The mutant PERK-ΔLD-KA does present all the phosphorylation sites for the S115/117/119 (shown in Fig EV3 C and A) but it does not rescue the PERK-mediated mitochondrial biogenesis phenotype, suggesting that the kinase activity is fundamental, probably together with the S115, 117 and 119. But the authors seem to ignore this and conclude "Collectively, our observations suggest that Ser715, Ser717 and/or Ser719 of PERK are indispensable for mitochondrial biogenesis and functions but not for the UPR".

Re: We the reviewer thank for the suggestion and revised the manuscript (P13 L6-7).

Fig. 4A: Student's t-test is the wrong statistical test for this kind of data with multiple comparisons /samples in one graph. The authors should use appropriated statistical tests like ANOVA in this case and through the manuscript. The authors use student's

t-test nearly ubiquitously but incongruently. Examples of misuse are in figures 4A and EV5D, but also why are data in Figure 5A analyzed using ANOVA, but an identical dataset in Fig EV5D with a t-test?

Re: Our response to this comment is described in the cover letter to the editor.

Fig. 4D: Also in this panel the authors show that both the PERK 3SA and the PERK KA mutants do not rescue GABPα transcriptional activity, indicating that the kinase activity of PERK is important. Hence the conclusion the author draw based on these results: "suggesting that PERK specifically regulates the GABPα pathway as a result of PERK phosphorylation at Ser715, Ser717 and/or Ser719 during BA differentiation." should be revisited to include the possibility that also the kinase activity

Re: We agree with the reviewer's suggestion and revised the manuscript (P14 L20-23).

-A major conclusion from the data of Fig. 4 is that PERK phosphorylation/kinase activity is mechanistically linked to GABPα activation, which in turn regulates the expression of mitochondrial genes and mito respiration, but it would be important to show that GABPα regulates mito cristae morphology. It remains to be clarified how does PERK cause the activation of GABPα. Why does it need its kinase activity -if it is not UPR dependent- and why does it need to be phosphorylated?

Re: We thank the reviewer for the suggestion. We are currently trying to identify the mechanism by which PERK activates the GABPα pathway, but unfortunately, we have not succeeded yet. To address this important point, investigation must be continued. Our revised manuscript has been toned down as described in the cover letter to the editor.

Fig. 5A: The authors should also include data showing the effects of the kinase dead

mutant (PERK- Δ LD-KA) on the thermogenic defect. Actually, since the PERK- Δ LD-3SA mutant only partially rescues it, again here there exists the possibility that PERK kinase activity is involved. Moreover, why doesn't the mutant PERK- Δ LD rescue completely the phenotype?

Re: Our response to this comment is described in the cover letter to the editor.

Minor points:

2C: Authors need more data to claim the lipid droplet content is unchanged. At least some microscopic staining should be shown.

Re: We agree with the reviewer's concern, although Oil Red O staining is used for staining of neutral triglycerides and lipids and measurement of their amounts in cells. We have shown a larger field of view in the electron micrographs of PERK-knockdown BAs (revised Fig 3C, described in P10 L2-3).

EV2C: in the same blot please also show the increased P-PERK (taking along also the condition of 6H and 12h of medium incubation where in the figure 2A you show P-PERK) and a positive control of TG/TM at different time points especially because ATF4 signal is not that clear here. If possible eIF2a-P /eIF2a levels should be shown as well, as eIF2a is a direct target of PERK.

Re: We thank the reviewer for the suggestion. The band shift and autophosphorylation of PERK at detailed time points during differentiation have been shown, as well as positive thapsigargin controls at 1 and 6 h (revised Fig 2A). Phosphorylation of eIF2 α was observed during BA differentiation, and PERK deficiency did not affect it at all (data not shown). Moreover, translational attenuation was not observed at 12 and 24 h on day 2 (revised Fig 2D). We do not yet have an answer for the discrepancy between PERK-independent eIF2 α

phosphorylation and translational progress. Since these data may confuse our conclusion, we did not include the eIF2 α phosphorylation data in our manuscript.

EV2G-I: The immunostaining of the mitochondria and the quantification need improvements.

Re: We thank the reviewer for the suggestion. To solidify our conclusion that PERK deficiency has no effect on the areas of the ER and mitochondria, we have added quantification data from electron micrographs (revised Fig S3F and G).

Reviewer #3:

Kato et al. propose that PERK is essential for BAT thermogenesis. However, this hypothesis is not adequately supported by the data provided.

1. Regulation of PERK phosphorylation during brown adipocyte differentiation is weak. The authors only observe a modest increase in PERK phosphorylation during a short period of the differentiation process (after 6 and 12 hours on day 2). Furthermore, it is unclear whether this increase is actually attributable to the differentiation process or is simply due to the fact that the medium was changed, thereby replenishing various nutrients and growth factors. In the absence of robust regulation of PERK phosphorylation during brown adipocyte differentiation, it is unclear whether PERK phosphorylation is truly relevant in this context or if manipulating PERK levels simply has non-specific effects on mitochondria.

Re: Our response to this comment is described in the cover letter to the editor.

2. The proposed mechanism is also inconsistent with the data provided. The authors show that ER-mitochondria contacts are increased in differentiated brown adipocytes and hypothesize that these contacts are important. However, the authors then attribute the effects of PERK to a transcriptional pathway involving GABP α , which is not clearly linked to the ER-mitochondria contact observation. Similarly, the authors propose that PERK regulates mitochondrial biogenesis through GABP α , but then show that PERK knockdown has no effect on mitochondrial area or DNA content. The major defect appears to be in the morphology of the cristae; however, the mechanism responsible for this effect is not explored experimentally, only conceptually in the discussion.

Re: Our response to this comment is described in the cover letter to the editor.

3. Lastly, the in vivo data are unconvincing. PERK is a ubiquitously expressed and generally important protein. A global, developmental PERK knockout mouse is thus not a reliable model for studying intrinsic PERK function in BAT. The body surface temperature difference is also very modest.

Re: We fully agree with the reviewer's concern. As we have described in the manuscript (P15 L23-P16 L1), adult *PERK*^{-/-} mice have the phenotype of hyperglycaemia due to progressive degeneration of pancreatic secretory cells. BAT-specific *PERK*-3SA knock-in mice are better for *in vivo* analysis because of the lack of UPR disturbance; however, unfortunately, we have not yet succeeded in this endeavour. Investigation should be continued.

Editor:

We thank editor for the thoughtful comments and advice. Our manuscript has been revised by including additional experimental data according to the editor and reviewers' suggestion.

Point-by-point responses to the reviewers' comments suggested by the editor.

Shaded texts with blue and italic Arial characters are the editor's comments.

Texts with blue and italic Arial characters are the reviewers' comments.

Texts with black Times New Roman characters are our responses.

The reviewers appreciated your data, but would have expected more mechanistic insight into how PERK is activated and how phosphorylation of PERK signals to GABPalpha. Addressing this criticism is not needed for publication in Life Science Alliance, and we would thus like to invite you to submit a revised version of your manuscript, addressing the technical issues raised and those that pertain to lack of support for the existing data. More specifically, please provide a point-by-point response and edit the text / modify data representation according to the referees' requests. In addition, quantification and proper statistical analysis of all experiments are required.

Re: We thank you for your thoughtful comments. We understand the importance of quantification and have performed careful statistical analyses. With regard to organellar changes, the perimeters of mitochondria and the ER and the lengths of ER-mitochondria contact sites were quantified from electron micrographs and analysed by Student's *t-test* (revised Fig 1C, D, L-N and S3F-J).

In our previous data on ATP production, we showed the ratios of glycolysis-dependent ATP production and OXPHOS-dependent ATP production as fold differences relative to the total amount of intracellular ATP on day 0 (previous Fig 1B, 2I, 3D, 4E, EV4A and EV4H). However, the total amounts of intracellular ATP were different in different stages of brown preadipocytes and BAs (revised Fig 1E and 4D). Therefore, the ratio of OXPHOS-dependent ATP production

to total ATP production in each cell stage would be a better parameter for assessment of changes in OXPHOS-dependent ATP production capacity during BA differentiation (revised Fig 1F, 4E, 5G, 6G and S5F). These data were analysed by Student's *t-test*.

We agree with the comment from reviewer #2 related to the previous Fig 4A. Since the expression of each mRNA in PERK-knockdown cells is shown relative to that in control cells, the relative mRNA expression levels of each gene should be shown as independent graphs, as shown in the revised Fig 6A. The data in each graph were statistically analysed by Student's *t-test* (revised Fig 6A). Following this reviewer's comment, the data in previous Fig EV5D were statistically analysed by ANOVA (revised Fig S6D). Related to this suggestion, the data in revised Fig 3B were statistically analysed by ANOVA.

The newly added data in the revised manuscript were analysed statistically (Fig 1C-I, 1L-N, 4F-H, S3B-D and S3F-J).

Please also address the following specific reviewer points:

Reviewer #2 point 3:

There should be a more detailed quantification in terms of the organellar changes, which seem drastic (why is the ER so dramatically reduced?), usually ER-mitochondria contact length should be normalized to mitochondrial perimeters (or number). It would be also important to see a bigger field of view of the EM, not just a zoom. How does the overall cell look like? Beyond morphology, do the mitochondria change in number and subcellular distribution?

Re: We fully agree with the reviewer's concern. We carefully measured the changes in ER and mitochondrial perimeters and ER-mitochondria contact lengths during differentiation from day 0 to day 6. The ER-mitochondria contact sites are presented as the measured values and as ratios normalized to the ER and mitochondria perimeters (revised Fig 1L-N). The ER perimeter was significantly reduced (revised Fig 1D), and the ratio of contact sites to the ER perimeter was significantly increased (revised Fig 1M). Furthermore, the mitochondrial perimeter was

strongly increased (revised Fig 1C), and the ratio of contact sites to the mitochondrial perimeter was also increased (revised Fig 1N). These points are described on P7 L17-20 and P8 L13-21. We also investigated the effects of PERK deficiency on the areas and perimeters of mitochondria and the ER and on the ER–mitochondria contact sites (revised Fig S3A-C and E-J). We statistically analysed the data and found that there were no significant differences between control and PERK-deficient cells (described on P10 L7-12).

We agree with the importance of showing a larger field of view for the electron micrographs. As shown in revised Fig S1, a few lipid droplets (LDs) were observed on day 4, and multiple LDs emerged on day 6. At these stages, the mitochondria were clearly increased in number and distributed throughout the cells (revised Fig S1, days 4 and 6), and expanded mitochondria were observed on day 6 (revised Fig 1B). These points are described on P7 L15-20. We also show larger fields of view for the electron micrographs in PERK-knockdown BAs (revised Fig 3C, described on P10 L2-3).

- reviewers' concerns pertaining to inconsistencies between the different datasets

Reviewer #2:

-Can the authors also explain why in Fig 2F, the ER associated proteins -especially GRP94- show no changes till day 6, while in Fig. 1D they are reduced already at day 2, leading the authors to conclude that there is a reduced ER area.

Re: We agree with the reviewers' concern about the discrepancies in the GRP94 expression data in previous Fig 1D and previous Fig 2F. We performed these experiments several times. Compared with that of GRP78 (BiP), the expression level of GRP94 seems to be easily altered by the cell culture conditions. Although we have no specific answer yet, the cells in previous Fig 2F may have been under stress conditions because of siRNA transfection. We have thus omitted the data for GRP94, since we believe the data sets for GRP78 and HERP (ER stress markers) are sufficient to reach our conclusion (revised Fig 1K and 3F). Moreover, we have added the Sec61 α expression data to the revised Fig 3F, as shown in the previous Fig 1D

(revised Fig 1K).

- address Point #2 and #3 from referee #1

Reviewer #1 point 2:

2) The claim that PERK is physiologically phosphorylated independently of ER stress / UPR should be more carefully investigated. Basic signaling markers of the pathway should be investigated such as the ER stress sensor activity, the transcription factors regulated and some key UPR gene targets. The authors assayed PERK phosphorylation, ATF6 activity (indirectly through GRP78), and ATF4 (Chop and Gadd34) induction. However, for IRE1a signaling the analysis is incomplete. Firstly, the authors use a total IRE1a antibody to comment on the phosphorylation status of IRE1a; secondly, the authors do not investigate downstream targets of the IRE1a pathway (e.g. Xbp1 splicing). A careful investigation of all UPR signaling pathways is necessary to confirm that PERK is specifically activated independently of other UPR pathways, which is a major claim of the manuscript.

Re: We agree with the reviewer's suggestions to investigate three UPR signalling pathways carefully. We carefully analysed the shift in the molecular weight of IRE1 α and the expression of spliced XBP1 (XBP1[S]) at detailed time points from day 2 to 4. Changes in both the electrophoresis pattern of IRE1 α and the expression of XBP1(S) were observed after treatment with the ER stressor thapsigargin but not after replacement of the medium with differentiation enhancement medium (revised Fig 2A). In the same lysate, PERK activation, which was monitored by assessment of PERK autophosphorylation at Thr980, was examined, and PERK was found to be temporarily activated at 6 and 12 h after the enhancement of BA differentiation (revised Fig 2A). These points are described on P9 L6-10.

Reviewer #1 point 3:

3) The authors state that PERK was temporarily phosphorylated after 6h to 12h of

culture with differentiation enhancement medium on day 2. How can the authors be sure that this phosphorylation is not induced by the medium change itself which causes cellular stress? The authors should perform a detailed time course throughout the BA differentiation to make sure the temporal phosphorylation is not induced by media change per se.

Re: We fully agree with the reviewer's concern and performed a control experiment by changing the medium to a medium with the same composition. The molecular weight shift and autophosphorylation of PERK were not observed to be caused by the media change per se (revised Fig 2A).

- determine whether increased ATP production through OXPHOS is significant and measure oxygen consumption rate (referee #2)

Reviewer #2 Fig. 1A: Panel B:

Fig. 1A: Panel B: Is the increased ATP production through OXPHOS significant? Also, it would be mandatory to couple this indirect measurement of mitochondrial ATP production, to actual measurements of oxygen consumption rate.

Re: We thank the reviewer for the suggestion. We noticed that the graph in previous Fig 1B was not suitable for presentation of the increased ratio of OXPHOS-dependent ATP production because the total amount of ATP differed between brown preadipocytes and differentiated BAs (revised Fig 1E). Thus, we have revised previous graphs to graphs that show the ratio of OXPHOS-dependent ATP production to the total amount of intracellular ATP amount in each stage of cells and analysed the results statistically (revised Fig 1F, 4E, 5G, 6G and S5A and F).

We also measured the oxygen consumption rate (OCR) by using Seahorse XF Analyzer. Not only basal OCR but also extracellular acidification rate (ECAR), which represent glycolytic pathway activity, were significantly increased during BA differentiation (revised Fig 1G and H). However, since the increase in the rate of basal OCR during BA differentiation was higher than

that of ECAR, the basal OCR/ECAR ratio was significantly higher in BAs than in brown preadipocytes (revised Fig 1I). These points are described on P7 L20-P8 L5. Moreover, we measured basal OCR and ECAR in PERK-knockdown BAs and found that basal OCR, but not ECAR, was reduced by PERK deficiency (revised Fig 4F-H). These points are described on P11 L8-9.

- use commercially available anti-PERK antibody to test that autophosphorylation is evoked during differentiation (referee #2)

Reviewer #2 Fig. 2A:

Fig. 2A implies that PERK phosphorylation causes the shift in the molecular weight of the protein. Since these are murine cells, the authors should use anti-P-PERK antibody (commercially available and validated) to analyze whether the autophosphorylation site, and therefore the active form of PERK, is evoked during differentiation. Since the use of PERK kinase dead mutant highlight that the PERK kinase activity is required for many (if not all) the effects observed, this analysis is required.

Re: We thank the reviewer for the thoughtful comment. As described above in our response to “Reviewer #1 point 2”, we found that PERK was temporarily autophosphorylated at 6 and 12 h after the enhancement of BA differentiation (revised Fig 2A, described on P9 L6-10). As the reviewer comments, PERK kinase activity is required for the expression of mitochondrial proteins (revised Fig 5E), mitochondrial function (revised Fig 5F and G) and transcriptional activation of GABP α (revised Fig 6F). Taken together, our data suggest that both kinase activity and phosphorylation at Ser715, Ser717 and/or Ser719 of PERK during BA differentiation are required for mitochondrial function. These points are commented on the revised manuscript (P12 L20-P13 L1, P14 L20-23 and P16 L19-22).

- address point #1 from referee #3

Reviewer #3 point 1:

1. Regulation of PERK phosphorylation during brown adipocyte differentiation is weak. The authors only observe a modest increase in PERK phosphorylation during a short period of the differentiation process (after 6 and 12 hours on day 2). Furthermore, it is unclear whether this increase is actually attributable to the differentiation process or is simply due to the fact that the medium was changed, thereby replenishing various nutrients and growth factors. In the absence of robust regulation of PERK phosphorylation during brown adipocyte differentiation, it is unclear whether PERK phosphorylation is truly relevant in this context or if manipulating PERK levels simply has non-specific effects on mitochondria.

Re: We thank the reviewer for the comment. As described above in our response to “Reviewer #1 point 3”, we carefully examined whether PERK is phosphorylated during the enhancement of BA differentiation by performing a control medium change experiment. The shift in the molecular weight of PERK was observed only at 6 and 12 h after replacement of the medium with differentiation enhancement medium and was not caused by the medium change per se (revised Fig 2A, described on P9 L3-8). Since BA differentiation is mediated by many signalling pathways other than the PERK pathway, it may be difficult to investigate the sufficiency of 3S phosphorylation for mitochondrial inner membrane protein biogenesis and function. Thus, we have toned down our discussion and have described only the requirement of PERK itself or PERK-3S in our revised manuscript (e.g., P6 L24-P7 L2 and P16 L19-22).

toning down the conclusions

I'd advise to change the text to address the following reviewer comments: Rev#1, point 1 and 5; Rev#2, point regarding Fig2D, point regarding lack of rescue with PERK-deltaLD-KA; Rev#3, point 2

Reviewer #1 point 1:

1) There are several claims and speculations in the manuscript that are not sufficiently supported. E.g. at the end of the introduction the authors state that "the molecular

mechanism of mitochondrial stress-induced PERK activation may be a target through which to increase energy metabolism without affecting ER homeostasis." The authors write "Identification of the kinase that is activated by mitochondrial stress and contributes to PERK phosphorylation may clarify the detailed mechanism by which mitochondrial biogenesis is regulated in BAs." What is the evidence that it is mitochondrial stress activates the PERK-GABPa axis? It might as well be a non-mitochondrial signal. Similarly, the involvement of PGC1a in mediating the response is also purely speculative.

Re: We thank you for the suggestion and agree that some sentences were overwritten without clear evidence. Although we have shown that CCCP (a mitochondrial oxidative phosphorylation uncoupler) induced phosphorylation of PERK-3S in BAs (revised Fig S6C), there is no evidence that the phosphorylation of PERK-3S during BA differentiation is triggered by mitochondrial stress. Therefore, we have removed "mitochondrial stress" and have toned down the two sentences in the revised manuscript as follows:

P18 L4-6

"Identification of the kinase that is activated by mitochondrial stress and contributes to PERK phosphorylation may clarify the detailed mechanism by which mitochondrial biogenesis is regulated in BAs." > "Identification of the kinase that is activated during BA differentiation and contributes to PERK phosphorylation may clarify the detailed mechanism by which mitochondrial inner membrane protein biogenesis is regulated in BAs."

P19 L20-23

"Although further investigation is needed to clarify the mechanisms by which PERK is phosphorylated at Ser715, Ser717 and/or Ser719 during BA differentiation, the molecular mechanism of mitochondrial stress-induced PERK activation may be a target through which to increase energy metabolism without affecting ER homeostasis." > "Although further investigation is needed to clarify the mechanisms by which PERK is phosphorylated at Ser715, Ser717 and/or Ser719 during BA differentiation, PERK-GABPa axis signalling may be a target

through which to increase energy metabolism without affecting ER homeostasis.”

We also agree that the contribution of PGC1-1 α is just a possibility. The revised manuscript has been toned down as follows:

P18 L21-24

“The reductions in Cyt C, COX4 and UCP1 in PERK-deficient BAs were partially, but not completely, ameliorated by exogenously expressed PGC-1 α (Fig EV5E). Some of the GABP α target mitochondrial gene regulation mediated by PERK might be contributed by mechanisms other than PGC-1 α .” > “Since the reductions in Cyt C, COX4 and UCP1 in PERK-deficient BAs were partially ameliorated by exogenously expressed PGC-1 α (Fig S6E, lane 6), some mitochondrial genes targeted by PERK may be regulated by the cooperation of PGC-1 α with GABP α .”

Reviewer #1 point 5:

5) The observation that only some mitochondria possessed abnormal cristae in PERK-deficient brown adipocytes is interesting. Do the mitochondria with abnormal cristae have more contact sites to ER than mitochondria with a "normal" morphology in PERK-deficient BAs?

Re: We thank the reviewer for the comment. Since we have no data on the difference between the ER-contact area on mitochondria with abnormal cristae and that on mitochondria with dense parallel cristae, our discussion has been toned down, and we have revised manuscript by adding the following sentence:

P18 L7-13

“Another important question is how PERK regulates mitochondrial inner membrane protein biogenesis and crista formation. Although PERK deficiency had no effect on the areas of ER-mitochondria contact sites (Fig. S3H-J), PERK enrichment in the MAM might contribute specifically to parallel crista formation in ER-attached mitochondria (Verfaillie et al, 2012). Since we do not yet have evidence regarding the physiological relevance of the increased areas

of ER-mitochondria contact sites to PERK-GABP α axis-mediated mitochondrial function, further investigation is necessary."

Reviewer #2 Fig 2D:

Fig. 2D: In the EM analysis the authors show in this panel, clear differences in the size and elongated morphology of the mitochondria, beyond crista differences, are visible in the PERK silenced cells (quite visible also in EV2). Moreover, in 2F silencing PERK seems to impact all mitochondria proteins including tom20, can the authors comment on this and perhaps expand the analysis to other outer mitochondria proteins. Without a clear explanation of these effects the following conclusion; "These observations are consistent with the result that PERK deficiency affects the number of cristae but not the total area of mitochondria (Figs 2E and EV2I)" is muddled.

Re: We understand the reviewer's concern. Mitochondrial perimeter was quantified from electron micrographs. As shown in revised Fig S3G, a marginal, but not significant, decrease in mitochondrial perimeter was observed in the PERK-deficient BAs. This might be a reason why the expression of Tom20 was slightly reduced in PERK-deficient BAs (revised Fig 3F, day 6). The possibility that PERK regulates mitochondrial outer membrane proteins cannot be ruled out; we have toned down the discussion and have described this possibility on P10 L18-24.

Reviewer #2 Fig 5A:

Fig. 5A: The authors should also include data showing the effects of the kinase dead mutant (PERK- Δ LD-KA) on the thermogenic defect. Actually, since the PERK- Δ LD-3SA mutant only partially rescues it, again here there exists the possibility that PERK kinase activity is involved. Moreover, why doesn't the mutant PERK- Δ LD rescue completely the phenotype?

Re: We understand the reviewer's concern. Since PERK- Δ LD-KA did not rescue the reduced

expression of UCP1 in PERK-deficient BAs at all (revised Fig 5E), PERK- Δ LD-KA may not rescue the defect of the β_3 AR stimulation-induced thermogenesis. The percentage of retroviruses expressing the GFP variant Venus in BAs was 67%. This may be a reason why the mutant PERK- Δ LD did not completely rescue the phenotype of PERK-deficient BAs. These points are described in the revised manuscript, and the discussion have been toned down (P15 L15-21).

Reviewer #3 point 2

2. The proposed mechanism is also inconsistent with the data provided. The authors show that ER-mitochondria contacts are increased in differentiated brown adipocytes and hypothesize that these contacts are important. However, the authors then attribute the effects of PERK to a transcriptional pathway involving GABP α , which is not clearly linked to the ER-mitochondria contact observation. Similarly, the authors propose that PERK regulates mitochondrial biogenesis through GABP α , but then show that PERK knockdown has no effect on mitochondrial area or DNA content. The major defect appears to be in the morphology of the cristae; however, the mechanism responsible for this effect is not explored experimentally, only conceptually in the discussion.

Re: We understand the reviewer's concern and fully agree that the relation between the increased ER-mitochondria contact and the PERK-GABP α -pathway mediated mitochondrial function is just a speculation. Disturbance of ER-mitochondria contact or exclusion of PERK from the MAM may enable us to verify this hypothesis, but we have not yet succeeded in these endeavours. Further investigation is necessary. We have toned down this point in the revised manuscript, as follows:

In Abstract (P4 L5-7)

“Here, we show the importance of an endoplasmic reticulum (ER)-mitochondria crosstalk signaling pathway mediated by the ER-resident sensor PKR-like ER kinase (PERK) in the thermogenesis of brown adipose tissue (BAT).” > “Here, we show the importance of the

endoplasmic reticulum (ER)-resident sensor PKR-like ER kinase (PERK) in the mitochondrial thermogenesis of brown adipose tissue (BAT).”

In Introduction (P7 L2-4)

“Overall, our data strongly suggest that a ER-mitochondria crosstalk mediated by PERK is indispensable for mitochondrial biogenesis and thermogenesis in BAT.” > “Overall, our data suggest that the activation of the PERK-GABP α pathway during BA differentiation is indispensable for mitochondrial inner membrane protein biogenesis and thermogenesis in BAT.”

In Result (P9 L1)

“We next examined the involvement of the UPR signaling pathway in BAs that had acquired expanded ER-mitochondria contact sites.” > “We next examined the involvement of the UPR signalling pathway in BAs.”

Addition the following sentence into Discussion (P18 L10-13)

“Since we do not yet have evidence regarding the physiological relevance of the increased areas of ER-mitochondria contact sites to PERK-GABP α axis-mediated mitochondrial function, further investigation is necessary.”

January 21, 2020

RE: Life Science Alliance Manuscript #LSA-2019-00576-TR

Prof. Hideki Nishitoh
University of Miyazaki
5200, Kihara, Kiyotake
Miyazaki 8891601
Japan

Dear Dr. Nishitoh,

Thank you for submitting your revised manuscript entitled "ER-resident sensor PERK is essential for mitochondrial thermogenesis in brown adipose tissue". One of the original reviewers re-evaluated your work and your response to the original concerns in light of the revision requests we made. I am glad to say that the reviewer appreciates the changes introduced in revision. We would thus be happy to publish your paper in Life Science Alliance pending final revisions necessary to meet our formatting guidelines:

- Please note that figures can only span a single page. Please revise Fig 1, 5 and 6 accordingly (you can introduce more figures by splitting these into several ones)
- Please upload all figures, including supplementary figures, as individual files. All figure legends should get provided in the main manuscript docx file
- Please add scale bars to Fig. 4A and S3K

A. FINAL FILES:

B. MANUSCRIPT ORGANIZATION AND FORMATTING:

Sincerely,

Reviewer #1 (Comments to the Authors (Required)):

In this revised version of their MS the authors have addressed some of the reviewer's concern properly and toned down some of the conclusions that were too speculative.

January 23, 2020

RE: Life Science Alliance Manuscript #LSA-2019-00576-TRR

Prof. Hideki Nishitoh
University of Miyazaki
5200, Kihara, Kiyotake
Miyazaki 8891601
Japan

Dear Dr. Nishitoh,

Thank you for submitting your Research Article entitled "ER-resident sensor PERK is essential for mitochondrial thermogenesis in brown adipose tissue". It is a pleasure to let you know that your manuscript is now accepted for publication in Life Science Alliance. Congratulations on this interesting work.

DISTRIBUTION OF MATERIALS:

Again, congratulations on a very nice paper. I hope you found the review process to be constructive and are pleased with how the manuscript was handled editorially. We look forward to future exciting submissions from your lab.

Sincerely,
